# Worst-case Few-shot Evaluation: Are Neural Networks Robust Few-shot Learners?

## Abstract

Neural networks have achieved remarkable performance on various few-shot tasks. However, recent studies reveal that existing few-shot models often exploit the spurious correlations between training and test sets, achieving a high performance that is hard to generalize. Motivated by a fact that a robust few-shot learner should accurately classify data given any valid training set, we consider a worst-case few-shot evaluation that computes worst-case generalization errors by constructing a challenging few-shot set. Specifically, we search for the label-balanced subset of a full-size training set that results in the largest expected risks. Since the search space is enormous, we propose an efficient method NMMD-attack to optimize the target by maximizing NMMD distance (maximum mean discrepancy based on neural tangent kernel). Experiments show that NMMD-attack can successfully attack various architectures. The large gap between average performance and worst-case performance shows that neural networks still suffer from poor robustness. We appeal to more worst-case benchmarks for better robust few-shot evaluation.

## 1 Introduction

Given a limited number of supervised samples, few-shot learning aims to achieve high generalization performance on unseen test data (Wang et al., 2020; Yue et al., 2020). Recent years have witnessed rapid advancement of few-shot learning, particularly with neural networks pre-trained with self-supervised data (Brown et al., 2020; Dosovitskiy et al., 2021; Tan & Le, 2021). Neural networks gradually become the dominant solution to few-shot learning. Some networks (He et al., 2021; Xu et al., 2022) even surpass humans in standard benchmarks (Wang et al., 2019; Mukherjee et al., 2021).

Despite promising results on fixed sets, recent work (Sagawa et al., 2020; Taori et al., 2020a; Koh et al., 2021; Tang et al., 2022) reveals that neural networks as few-shot learners exacerbate spurious correlations and easily fail on distribution shifts. Spurious correlation is a typical representation learning problem in which models learn to classify based on superficial features. For example, when learning the class label *pigs* from a few pig images, neural network models sometimes learn to guess based on superficial features (e.g., background with farm fences) rather than learn to generalize base on essential features (e.g., the facial characteristics of pigs), as shown in Figure 1.

Over-fitting to spurious attributes brings performance increase hallucination but does not guarantee better robustness, which explains over-optimistic performance on existing benchmarks (Mutton et al., 2007; Vinyals et al., 2016; Oreshkin et al., 2018; Schick & Schütze, 2021; Alayrac et al., 2022a). The performance of models are assessed according to the averaged test accuracy given a fixed training set (1-fold evaluation) or several random subsets of the training set (k-fold evaluation). In that procedure, it is easy for the superficial features to be carried by the few-shot sets and eventually exploited by the neural networks since training and test data usually come from the same data distribution in the construction of the benchmark (Sagawa et al., 2020).

Motivated by a fact that a robust few-shot learner should accurately classify data given any valid training set, we propose a *worst-case* evaluation for few-shot learners in this work. Worst-case evaluation targets to evaluate generalization error bounds. Instead of randomly sampling few-shot sets, we search for the worst-case few-shot set from a full-size training set with balanced labels. An illustration of worst-case few-shot evaluation is shown in Figure 1. Inspired by the notion that spurious correlations often arise from common statistical features (Sagawa et al., 2019; Tang et al., 2022), the distribution of unbiased samples generally has large divergence with the full-size training

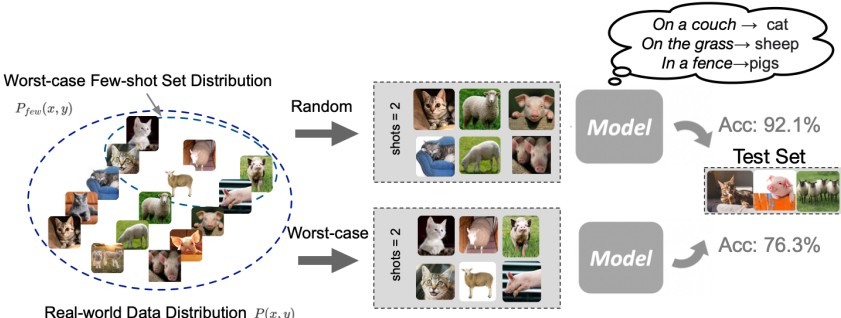

Figure 1: An illustration of worst-case evaluation. Spurious correlations affect few-shot evaluation. Since training and test data usually come from the same distribution, over-fitting to spurious features brings performance increase but does not bring better robustness. For example, models learn to classify pig based on superficial features (e.g., in a fence), rather than the shape. In this work, we consider a worst case evaluation for few-shot robustness evaluation by extracting a challenging and label-balanced subset from a full-size training set with the largest expected risk.

set. Therefore, we adopt a distribution divergence maximization approach NMMD-attack to find the most challenging few-shot set. This approach is also theoretically guaranteed as the generalization error of a few-shot set is bounded by the maximum mean discrepancy (MMD) distance (Gretton et al., 2012) between the few-shot distribution and the original training distribution. The goal of maximizing generalization error can then be simplified to maximizing the MMD distance between the few-shot set and the full-size training set. Following the MMD maximization principle, we use the MMD distance in the hypothesis space to define group (or set) distance. Since the MMD distance is still intractable, we borrow neural tangent kernel, an approximation to over-parameterized neural networks, to estimate MMD distance without optimization. Given the searched subset, we train models and report test accuracy.

Experiments show that NMMD-attack challenges high few-shot performance in randomly-sampled cases. It can successfully attack various model architectures with large performance drops. For example, the performance of DenseNet-121 drops by 10.02% on the generated few-shot set. Also, our case study on ImageNet-1K and CIFAR-10 demonstrates that the generated few-shot sets show much fewer spurious attributes than randomly-sampled few-shot training sets.

This work re-examine the actual ability of representative neural networks on few-shot cases. The large performance drop indicates large improvement space in future work. Actually, the bias to spurious attributes can be a severe system bug and loophole for all few-shot learners. The attacker can manipulate sets with unseen correlations to destroy a model, which is hard to detect. How to avoid learning spurious features is a hopeful direction to improve few-shot robustness. Furthermore, compared with existing few-shot benchmarks, worst-case evaluation can provide a new view demonstrating how worse a model can be such that we can prepare backup plans in case of accidents in real-word applications. In this work, we provide a feasible solution to estimate generalization error bounds for few-shot cases, we appeal to more worst-case benchmarks for better few-shot evaluation in future.

## 2 RELATED WORK

In this work, we review related topics, including adversarial attack, distribution shift, and distribution robustness optimization.

Our work comes as a form of attack inspired by adversarial attack literature (Szegedy et al., 2013; Goodfellow et al., 2014). Adversarial attacks aim to fool neural networks while keeping innocuous to humans. This form of attack, though effective, alters each sample independently and ignores group correlations. Performing attacks mainly concerns making slight variations, e.g. adding noise to the sample. Since few-shot cases usually have serious spurious correlations between groups, the target of worst-case few-shot evaluation is to evaluate robustness to distribution shifts.

Distribution shift naturally arises from real-world applications and has been widely used in robustness (Sagawa et al., 2019; 2021). Also, distribution discrepancy metrics, similar to the NMMD metric in our work, have been extensively used (Peng et al., 2019; Chen et al., 2020; Zhao et al., 2021) in predicting generalization performance under data shifts. Distributional robustness (Sinha et al., 2017) , though inspired by distribution shift, is a distinct direction. Distribution robustness requires models to minimize empirical error under any valid training set. Unlike attack methods (Madry et al., 2017; Cohen et al., 2019) that define their perturbation set with an $L_p$-ball, distributional robustness analyzes set robustness.

Other works have also explored distribution shifts and adopted samples with spurious correlations as benchmarks. However, most studies (Sagawa et al., 2019; Taori et al., 2020b; Sagawa et al., 2021) rely on prior knowledge of spurious correlations, which limits its application to real-world tasks with complicated and unseen correlations. Existing benchmarks for few-shot learning, like Mini-Imagenet (Dhillon et al., 2019), CIFAR-FS (Bertinetto et al., 2018), Tiered-ImageNet (Ren et al., 2018), mainly focus on cross-class query and label shifts. A more practical setting is based on the idea of "in context" learning few-shot examples for large pre-trained models (Alayrac et al., 2022b; Yang et al., 2022; Tsimpoukelli et al., 2021) and is the field that suffers over-optimistic performances. Our few-shot robustness evaluation also follows this setting. We are the first to appeal to worst-case benchmarks.

## 3 NOTATIONS

For a better description, we introduce the notations used in this paper. Let $\mathcal{X}$ be the data space, $\mathcal{Y}$ be the label space, and $\mathcal{H}$ be the hypothesis space in our investigation $\mathcal{H} : \mathcal{X} \to \mathcal{Y}$. We consider classification tasks for simplification. The training set $X$ consists of $n$ samples $(x_i, y_i)_{i=1}^n \sim P(X, Y)$. The model $f_\theta \in \mathcal{H}$ maps input to prediction $Y \in \mathbb{R}^d$, where $d$ denotes the dimensions of the output and $\theta$ represents the parameters of the model. To give a probabilistic view of the problem, the training distribution is denoted as $P(X, Y) = \frac{1}{n} \sum_{i=1}^n \delta_{(x_i, y_i)}$, where $\delta$ denotes the Dirac delta distribution concentrated on $(x_i, y_i)$. Consequently, $P(X) = \frac{1}{n} \sum_{i=1}^n \delta_{x_i}$ and $P(Y) = \frac{1}{n} \sum_{i=1}^n \delta_{y_i}$.

## 4 FEW-SHOT ROBUSTNESS: THE WORST-CASE GENERALIZATION ERROR

In this work, we discuss the problem of few-shot robustness. In traditional few-shot learning, we construct a few-shot set and measure the performance given the fixed training set and test set. Here we are interested in if different few-shot training sets lead to large model performance degradation. Inspired by the notion of robustness in distributional robust optimization (Sagawa et al., 2019; Kuhn et al., 2019), we aim to find a challenging few-shot training set $P_{\text{few}}$ that maximizes the expected risk on the test set. For convenience, we focus on a data-driven few-shot distribution, where $P_{\text{few}}$ is centered around empirical examples extracted from the original training set distribution.

We consider a few shot set with $k$ examples. The example index is defined as $I_k := \{i_1, i_2, ...i_k\} \subset [n]$, and therefore $P_{\text{few}}(X) = \frac{1}{k} \sum_{m=1}^k \delta_{X_{i_m}}$. For short, we also write $P_{\text{few}}(X)$ as $P_{I_k}(X)$. Let $f$ be the model hypothesis and $Q$ as the test distribution, the empirical risk of $f$ w.r.t training set $P$ is $\epsilon_P(f) = \mathbb{E}_{x \in P(x)} |\delta(f(x)) - P(y|x)|$, and the expected risk of $f$ w.r.t. $Q$ is denoted as $\epsilon_Q(f) = \mathbb{E}_{x \in Q(x)} |\delta(f(x)) - Q(y|x)|$. We assume the original training sets and test sets are sampled $i.i.d$ from a natural distribution. Then we can safely make Assumption. 4.1 that the full-size training set can teach a model with acceptable generalization errors.

**Assumption 4.1.** *(Generalization)* Let $\mathcal{H} : X \to Y$ be the hypothesis space, then there exists a model $f$ satisfies $f \in \mathcal{H}$ and $\epsilon_P(f) \leq \alpha$, $\epsilon_Q(f) \leq \beta$, where $\alpha, \beta$ are constants approaching zero.

In a real-world setting, a few-shot dataset is usually constructed by choosing a few representative samples (Parnami & Lee, 2022; Gao et al., 2020). Data imbalance is preferably avoided so as not to inject bias (Ochal et al., 2021; Wang et al., 2020). It is thus logical to assume that the few-shot subset should match the balanced label distribution of the original training set, as stated in Assumption. 4.2.

**Assumption 4.2.** *(Label Alignment)* Let $P_{I_k}(X, Y)$ be the few-shot set selected from $P(X, Y)$, then $I_k$ should satisfy $P(Y) = P_{\text{few}}(Y)$, i.e, $\frac{1}{n} \sum_{i=1}^n \delta_{Y_i} = \frac{1}{k} \sum_{m=1}^k \delta_{Y_{i_m}}$.

**Few-shot Robustness** Under these assumptions, the few-shot robustness problem we address asks to solve the worst-case problem denoted as,

$$\max_{I_k} \epsilon_Q(f_{I_k}), \tag{1}$$

where $f_{I_k}$ is a hypothesis that satisfies $\epsilon_{P_{I_k}}(f_{I_k}) \leq \alpha$, and $I_k$ satisfies $\frac{1}{n}\sum_{i=1}\delta_{Y_i} = \frac{1}{k}\sum\delta_{Y_{i,m}}, i \in [n], m \in I_k, I_k \subset [n]$. $\{P_{I_k}\}$ is the uncertainty set for constraining this distributional robustness problem. Different few-shot training sets $X_{I_k}$ lead to varied generalization errors.

Traditional few-shot evaluation usually adopts a fixed training set. Correlations between the training set and the test set largely decide the generalization errors on the test set. Higher performance does not represent better robustness due to the missing evaluation of the worst-case performance. To evaluate few-shot robustness in Eq. 1, the target is to find a subset $X_{I_k}$ to prevent models from learning biased features and increase the difficulty for generalization performance.

## 5 NMMD-ATTACK FOR WORST-CASE FEW-SHOT EVALUATION

This section introduces our method for evaluating the worst-case generalization error by constructing a challenging few-shot set based on Eq. 1. The intuition of NMMD-ATTACK is based on eliminating the spurious correlation between few-shot training set and test set. Since most samples hold spurious attributes (Sagawa et al., 2019) and samples with the same spurious attributes tend to be more similar, it is more likely to find less biased subsets based on distribution discrepancy. Therefore, we propose a NMMD-attack approach to find the worst-case few-shot subset with the largest MMD discrepancy from the original training distribution, while maintaining the balance of labels.

Here, We first proves the intuition that the worst-case can be achieved by maximizing the MMD distance between few-shot set and the original training set, and then introduce how to quantitatively evaluate the MMD distance using neural tangent kernel. Last, a practical solution is provided to generate the worst few-shot set. All proofs are available in the Appendix A.

### 5.1 FEW-SHOT ROBUSTNESS EVALUATION WITH MMD MAXIMIZATION

To find the worst-case few-shot set, our goal is to solve the optimization problem stated in Eq. 1. Intuitively, the generalization errors increase when there are fewer correlations and lower distribution similarity between the few-shot set and test set. In many benchmarks and contests, the test set is not directly available, but we can use a well-rounded training set as a resource of natural data distribution. In this case, our optimization goal becomes maximizing the distribution distance between the few-shot set and original training set. Here we use MMD distance following Gretton et al. (2012).

**Definition 5.1.** *(Distribution Distance)* Let $\mathcal{H}$ be the family of mappings from $\mathcal{X} \rightarrow \mathbb{R}^{\mathrm{d}}$, and let $P$ and $Q$ be two distribution, then the MMD distance can be defined as,

$$\mathrm{MMD}(H, P, Q) := \sup_{f \in \mathcal{H}}(E_{x \sim P}[f(x)] - E_{y \sim Q}[f(y)]). \tag{2}$$

We theoretically show that the MMD distance is an upperbound for the few-shot robustness metric.

**Theorem 5.1.** *(Few-shot Robustness Measured by MMD Discrepancy) Let $\mathcal{H}$ be the hypothesis space $X \rightarrow \mathbb{R}^{\mathrm{d}}$. $f_{I_k}$ is the empirical risk $\epsilon_{P_{I_k}}(f)$ minimizer, and $f$ is the hypothesis that minimizes expected risk $\epsilon_Q(f)$. Then*

$$\epsilon_Q(f_{I_k}) \leq \epsilon_Q(f) + MMD(P_{I_k}, P) + \epsilon_\alpha + t + \epsilon_{\mathcal{H}}, \tag{3}$$

*where $\epsilon_\alpha, t$ and $\epsilon_H$ are small constants describing the error occurred in training, the sampling behavior of training distribution and the hypothesis space complexity. Details are shown in Appendix A.1*

This theorem gives the intuition that a high discrepancy between the few-shot set and a training set results in high expected risk. Following statistics literature (Cheng & Xie, 2021), a biased empirical estimate of the MMD is obtained by replacing the population expectations with the empirical average computed on the samples. Then we have,

$$\mathrm{MMD}(\mathcal{H}, P, P_{I_k}) := \sup_{f \in \mathcal{H}}(\frac{1}{n}\sum_{i=1}^{n} f(x_i) - \frac{1}{m}\sum_{m=1}^{k} f(x_{i_m})) \tag{4}$$

For over-parameterized non-linear models, calculating MMD in the hypothesis space is intractable. Neural Tangent Kernel (NTK), however, is a simple approximation to understand neural networks as a kernel regression that can be optimized in its Reproducing Kernel Hilbert Space. Following the convention in NTK literature, we consider the NTK kernel function defined for optimization step $t \geq 0$ as:

$$K^t(x, x') := \langle \nabla_\theta f(\theta(t), x), \nabla_\theta f(\theta(t), x') \rangle. \tag{5}$$

Then the MMD distance could be deducted following kernel conventions (Gretton et al., 2012).

**Proposition 5.1.** *(NMMD Distance) The MMD static with NTK at time $t$ is,*

$$MMD_{K^t}^2(P, Q) = \int_{\mathcal{X}} \int_{\mathcal{X}'} K^t(x, x')(\hat{p} - \hat{p}')(x)(\hat{p} - \hat{p}')(x')dxdx', \tag{6}$$

*where $\mathcal{X}$ and $\mathcal{X}'$ denotes the support set for $P$ and $Q$.*

When the neural network approaches its infinite width limit, the NTK matrix stays constant during training (Arora et al., 2019; Du et al., 2019), i.e., $K^t$ equals to $K^0$. This property enables us to accurately estimate the MMD distance without costly training. Together with the approximate ability of NTK for neural network training (shown in Proposition. A.1), we show that NMMD can be used as an approximation to the MMD distance.

**Theorem 5.2.** *(NMMD Approximation) Let $MMD_f$ be the MMD distance calculated with model $f \in \mathcal{H}$ and let $MMD_K$ be the MMD static w.r.t. the NTK kernel $K$. Assume $f$ has taken $t$ steps of training at a learning rate of $\eta$, and $f$ is $L_f$-Lipschitz Continuous. Denote NTK at initialization as $K(x, x') = \langle \nabla_\theta f(\theta(0), x), \nabla_\theta f(\theta(0), x) \rangle$. If $k \leq \frac{n}{2}$, then the two MMD statistics satisfies:*

$$MMD_f(P, P_{I_k})) = O(t\eta \cdot MMD_K^2(P, P_{I_k})) \tag{7}$$

This theorem formally justifies using NTK kernel to calculate MMD distances. Compared with estimating MMD distance with neural networks, this method requires no training, and its calculation can be easily decomposed into kernel similarity $K(x_i, x_j)$ with the help of the kernel trick. The simplified distance would be,

$$\text{MMD}_K^2(P, P_{I_k}) = \frac{1}{n^2} \sum_{i,k}^{n} K(x_i, x_k) + \underbrace{(\frac{1}{k^2} - \frac{2}{nk}) \sum_{s,t}^{k} K(x_{i_s}, x_{i_t})}_{\text{intra-set kernel similarity}} - \underbrace{\frac{2}{nk} \sum_{i \in I_k, j \notin I_k} K(x_i, x_j)}_{\text{inter-set kernel similarity}},$$
$$\tag{8}$$

This distance would achieve its maximum while maximizing the intra-set kernel similarity of the few-shot set and minimizing the inter-set kernel similarity.

## 5.2 IMPLEMENTATION

We have simplified the task of finding the worst few-shot set to maximizing the NMMD distance between a few-shot distribution and the full-size training distribution, as shown in Eq. 8. However, there are several challenges in implementation: (1) The computational complexity for the NTK matrix on all training examples is $O(n^2)$, making it hard to scale to large-scale sets; (2) Optimizing the MMD distance requires solving a combinatorial problem of $O(\binom{n}{k})$. In our work, we address these problems by taking advantage of the sparsity of NTK matrix.

**Efficient Optimization of Eq. 8** In experiments, we discovered the NTK is usually a sparse, diagonally dominant matrix. As shown in Figure 2, the diagonal terms $K(x_i, x_i)$ are more than 10x larger than cross terms $K(x_i, x_j)$. In this way, we can discard cross terms for simplification in Eq. 8. $K(x_i, x_i)$ is actually gradient norm $\langle \nabla_\theta f(\theta, x_i), \nabla_\theta f(\theta, x_i) \rangle_{i \in [n]}$. Note that the gradient of model $f : \mathcal{X} \to \mathbb{R}^d$ is $\nabla_\theta f(\theta(t), x) \in \mathbb{R}_{p \times d}$, $p$ is the number of parameters in $f$. The calculation of $K(x_i, x_i)$ can be written as,

$$K(x_i, x_i) = \langle \nabla_\theta f(\theta, x), \nabla_\theta f(\theta, x) \rangle = \sum_{i=1}^{p} \sum_{j=1}^{d} \nabla_{\theta_i} f_j(x)^2. \tag{9}$$

---

**Algorithm 1** Details of NMMD-Attack.

---

**Input**: Full-size training set $X = (x_i, y_i)_{i=1}^n$, few-shot set size $k$, model $f$ with random parameters $\theta$, initial parameters $\theta_0$.
**Output**: The few-shot attack set.

1: initialize $f_\theta$ with $\theta_0$
2: **for** $i = 1 : n$ **do**
3: $\quad G_i \leftarrow \langle \nabla_\theta f_\theta(x_i), \nabla_\theta f_\theta(x_i) \rangle$
4: **end for**
5: Sort examples in X based on $G_i$
6: For each label, select top $k$ examples into the final few-shot set $X_{\text{few}}$ to optimize Eq. 10
7: **return** $X_{\text{few}}$

---

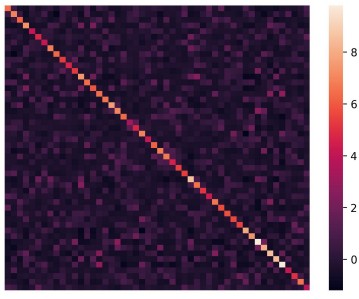

Figure 2: Visualization of NTK matrix with 50 examples randomly sampled from CIFAR-10.

After reducing the sparse terms in the NTK matrix, our new goal is to maximize the intra-set kernel similarity using the gradient norm terms:

$$\max_{I_k} \sum_{s=1}^{k} K(x_{i_s}, x_{i_s}), i_s \in I_k \tag{10}$$

**Implementation Algorithm**  Given Eq. 10, the target of finding the worst-case few-shot set is simplified into finding the few-shot set with the maximum sum gradient norms. The algorithm details are shown in Algorithm 1. First, We rank all training samples w.r.t. their gradient norms, i.e., $K(x_{j_1}, x_{j_1}) > K(x_{j_2}, x_{j_2}) > ... > K(x_{j_n}, x_{j_n})$, then we choose the top-k samples $(x_{j_s}, y_{j_s})_{s \in [k]}$ as our attack few-shot set. Last, we train models on the searched few-shot set and report its test accuracy.

## 6 EXPERIMENTS

**Datasets**  We explore the few-shot robustness on 4 image classification datasets, namely the CIFAR-10 dataset (Krizhevsky et al., 2009), the CIFAR-100 dataset (Krizhevsky et al., 2009), the MNIST dataset (LeCun et al., 1998) and the ILSVRC-2012 ImageNet dataset with 1K classes (Deng et al., 2009). We select a subset of the full-size training set as attack set. To build natural few-shot sets and guarantee the label alignment constraints, each label keeps $k$ corresponding instances in few-shot training set.

**Models and Hyper-parameters**  To evaluate the robustness of over-parameterized neural networks, we consider the following models: 1) **FFN**, a feed-forward neural networks; 2) **VGG** (Simonyan & Zisserman, 2014), a classical convolutional neural network; 3) **ResNet** (He et al., 2016), a residual neural network; 4) **ResNeXt** (Xie et al., 2017); 5) **DenseNet** (Huang et al., 2017). Besides, to verify the attack ability on pre-trained models, we also re-implement two pre-trained models: 1) Transformer-based **ViT** (Dosovitskiy et al., 2021) and 2) Convolutional-based **EfficientNetV2** (Tan & Le, 2021). Appendix B provides detailed model description and hyper-parameter settings. For each result, we conduct $m$ experiments and report the mean and variance. $m = 5$ for MNIST, CIFAR-10, and CIFAR-100, and $m = 3$ for ImageNet.

**Baseline and Comparison**  NMMD-attack finds the subset from the full-size training set to attack models. We implement an "average-case" baseline that randomly samples few-shot sets from the full-size training set as a comparison. After training models on the few-shot group, we report the accuracy of the test sets for both methods. To make a fair comparison, NMMD-attack and the baseline have the same dataset size and label distribution. To be specific, we extract 10% data from the training set as a few-shot set. Each label has 500 examples in MNIST, 500 examples in CIFAR-10, 50 examples in CIFAR-100, 100 examples in ImageNet-1K. Since NMMD-attack requires gradient norms, we implement various networks to calculate gradient norms. It needs to notice that such gradients do not require a trained network. Only architecture and initialization are required.

Table 1: The comparison between average performance and attack (FFN-attack) performance. "Acc." represents accuracy. "Test Acc. Gap" represents the gap between average performance and attack performance. "Abs." represents absolute gap and "Rel." represents relative gap. As we can see, NMMD-attack can successfully attack models with large performance drop.

| Datasets | Models | Average-case | | NMMD-attack | | Test Acc. Gap | |
|---|---|---|---|---|---|---|---|
| | | Train Acc. | Test Acc. | Train Acc. | Test Acc. | Abs. | Rel. |
| MNIST | FFN | 100.00 ±0.00 | 97.63 ±0.12 | 99.99 ±0.01 | **93.87** ±0.73 | 3.76 | 3.84 |
| | VGG-16 | 100.00 ±0.00 | 98.73 ±0.13 | 100.00 ±0.00 | **76.47** ±0.65 | 22.26 | 22.55 |
| | ResNet-18 | 100.00 ±0.00 | 98.65 ±0.05 | 100.00 ±0.00 | **75.41** ±0.54 | 23.24 | 23.55 |
| | ResNeXt-29 | 100.00 ±0.00 | 98.42 ±0.10 | 100.00 ±0.00 | **70.64** ±0.77 | 27.78 | 28.23 |
| | DenseNet-121 | 100.00 ±0.00 | 99.14 ±0.06 | 100.00 ±0.00 | **77.20** ±0.66 | 21.94 | 22.13 |
| CIFAR-10 | FFN | 100.00 ±0.00 | 49.38 ±0.47 | 100.00 ±0.00 | **42.29** ±0.73 | 7.09 | 14.36 |
| | VGG-16 | 98.85 ±0.31 | 66.20 ±0.91 | 98.82 ±0.84 | **51.43** ±1.07 | 14.77 | 22.31 |
| | ResNet-18 | 99.97 ±0.02 | 63.11 ±1.44 | 99.98 ±0.02 | **48.72** ±0.89 | 14.52 | 23.01 |
| | ResNeXt-29 | 99.52 ±0.48 | 61.75 ±0.58 | 99.66 ±0.16 | **48.59** ±0.62 | 13.16 | 21.31 |
| | DenseNet-121 | 99.53 ±0.45 | 71.19 ±0.87 | 99.64 ±0.18 | **54.76** ±1.56 | 16.43 | 23.08 |
| CIFAR-100 | FFN | 99.99 ±0.00 | 14.46 ±0.51 | 99.98 ±0.00 | **11.91** ±0.51 | 2.55 | 17.63 |
| | VGG-16 | 100.00 ±0.00 | 27.15 ±0.65 | 100.00 ±0.00 | **14.35** ±0.12 | 12.80 | 47.15 |
| | ResNet-18 | 100.00 ±0.00 | 24.91 ±0.20 | 100.00 ±0.00 | **14.27** ±0.29 | 10.64 | 42.71 |
| | ResNeXt-29 | 100.00 ±0.00 | 23.88 ±0.42 | 100.00 ±0.00 | **14.54** ±0.40 | 9.34 | 39.11 |
| | DenseNet-121 | 100.00 ±0.00 | 32.74 ±0.68 | 100.00 ±0.00 | **17.18** ±0.17 | 15.56 | 47.53 |
| ImageNet-1K | FFN | 99.94 ±0.01 | 5.06 ±0.31 | 99.77 ±0.00 | **3.14** ±0.24 | 2.08 | 51.49 |
| | VGG-16 | 98.11 ±0.02 | 14.22 ±0.24 | 97.97 ±0.05 | **9.74** ±0.50 | 2.54 | 40.71 |
| | ResNet-18 | 99.96 ±0.00 | 30.06 ±0.29 | 99.72 ±0.00 | **23.66** ±0.42 | 6.93 | 38.01 |
| | ResNeXt-50 | 99.94 ±0.01 | 37.97 ±0.09 | 99.66 ±0.05 | **26.51** ±0.50 | 9.72 | 37.28 |
| | DenseNet-121 | 99.84 ±0.01 | 37.76 ±0.73 | 99.61 ±0.02 | **28.49** ±0.36 | 10.02 | 37.63 |

## 6.1 WORST-CASE EVALUATION ON RANDOM-INITIALIZED ARCHITECTURES

**NMMD-attack successfully attack various models with large performance drop.** As Table 1 illustrates, NMMD-attack can successfully attack neural networks with worse generalization errors. For example, DenseNet-121 trained on the randomly sampling few-shot set achieves high test results with 71.19% on CIFAR-10, 99.14% on MNIST. When DenseNet-121 is trained on the adversarial few-shot sets generated by NMMD-attack , the performance drops significantly with 16.43% on CIFAR-10, 21.94% on MNIST, respectively.

**Neural Networks suffer from poor robustness.** First, Table 1 shows that the neural networks do not perform well on complicated few-shot datasets, including CIFAR-100 and ImageNet-1K with over 100 labels. It proves that few-shot generalization to complicated sets is still a big challenge. Second, even well-performing models on CIFAR-10 and MNIST easily learn spurious correlations and fail on adversarial sets. The large performance gap between NMMD-attack and average-case baseline implies that neural networks still suffer from the robustness problem.

**FFN-attack and ResNext-attack, two NMMD-attack variants, show better attack performance.** Since NTK requires gradient norm during the few-shot set search, we implement different networks as variants to observe how backbones affect attack performance. Due to the steady of NTK, we do not require a trained network during the search. The results are shown in Figure 3. FNN-attack and ResNeXt-attack achieve better attack performance. It indicates that gradients of FNN and ResNeXt give valuable information on unbiased examples. It also shows that worse models are more suitable for attack implementation. The over-fitting to spurious attributes empowers models with a solid ability to distinguish spurious attributes.

## 6.2 WORST-CASE EVALUATION ON PRE-TRAINED MODELS

We further verify the effectiveness of the NMMD-attack on pre-trained models. Recently, pre-training and fine-tuning have become a new paradigm of neural networks with strong learning ability (He et al., 2019). We conduct FFN-based NMMD-attack on two pre-trained models, transformer-based Vit-B/16 (Dosovitskiy et al., 2021) and convolution-based EfficientNetV2-S (Tan & Le, 2021) on CIFAR-10. These two models both are pre-trained on ImageNet-1K before downstream fine-tuning.

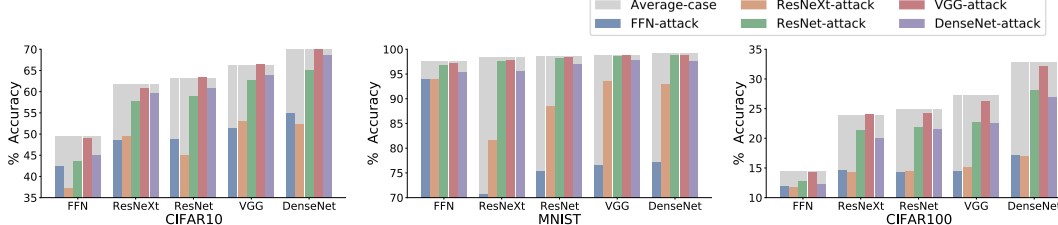

Figure 3: Results of NMMD-attack variants. Since NTK requires gradient norm during few-shot set search, we implement different networks. FNN-attack and ResNeXt-attack show good attack performance. Due to its simplicity, We adopt FNN-attack by default in the following experiments.

Table 2: The attack results (FFN-attack) on pre-trained models. The abbreviations follow Table 1. As we can see, pre-trained models have better defense performance on 500-shot cases but also suffer from poor robustness on 50-shot cases.

| Datasets | Models | Average-case | | NMMD-attack | | Test Acc. Gap | |
|---|---|---|---|---|---|---|---|
| | | Train Acc. | Test Acc. | Train Acc. | Test Acc. | Abs. | Rel. |
| CIFAR-10 (500-shot) | EffientNetV2-S | 99.72 ±0.03 | 91.71 ±0.29 | 99.75 ±0.04 | **91.50** ±0.43 | **0.21** | **0.23** |
| | ViT-B/16 | 100.00 ±0.00 | 97.14 ±0.14 | 100.00 ±0.00 | **96.97** ±0.13 | **0.17** | **0.18** |
| CIFAR-10 (50-shot) | EffientNetV2-S | 99.88 ±0.10 | 68.79 ±1.48 | 100.00 ±0.00 | **65.36** ±2.66 | **3.43** | **4.97** |
| | ViT-B/16 | 100.00 ±0.00 | 88.97 ±0.41 | 100.00 ±0.00 | **80.57** ±1.72 | **8.40** | **9.44** |

**Pre-trained networks outperform randomly-initialized networks under the same worst-case few-shot.** The first column in Table 2, CIFAR-10 (500-shot) shows performance of pre-trained networks on the few-shot set searched by FNN-attack. As we can see, pre-trained networks achieve much better robustness than randomly-initialized networks on 500-shot cases where the FFN-based NMMD-attack slightly deduces the few-shot learning ability of the pre-trained models. ViT-B/16 has 0.73% relative accuracy drop on CIFAR-10. It proves that large-scale pre-training can reduce the over-fitting to biased attributes and help robustness.

**Pre-trained networks suffer from poor robustness on smaller few-shot sets.** In particular, pre-trained models have solid performance on 500-shot cases. We are wondering if high performance can be generalized to smaller few-shot sets. We reduce the number of few-shot sets to 50-shot, where each label has 50 samples. As Table 2 illustrates, the FFN-based NMMD-attack consistently deduces few-shot performance on smaller few-shot sets. EfficientNetV2 has 4.97% relative accuracy drop and ViT-B/16 has 9.44% relative accuracy drop, respectively. In summary, NMMD-attack leads to performance deduction for various models, not only over-parameterized networks but also large pre-trained models. Moreover, we report our experiments for one-shot models in Appendix D.1. Our worst-case evaluation poses a higher challenge to defense such attacks for few-shot learning.

## 7 DISCUSSION

### 7.1 HOW IS THE QUALITY OF THE ADVERSARIAL FEW-SHOT SETS?

We first explore the instance similarity in the worst-case few-shot sets generated by NMMD-attack. To this end, we analyze the instance similarity via three metrics: 1) Structural Similarity (SSIM), a widely recognized method for measuring the similarity between two images (Wang et al., 2004); 2) Mutual Information (MI), a measure between two variables in information theory; 3)Peak signal-to-noise ratio (PSNR), a measure between super resolved image and the original one. Details of similarity measures can be found at Appendix C.

**The searched few-shot sets are natural and diverse.** Table 3 shows the instance similarity in the worst-case few-shot sets and random few-shot sets. As we can see, the searched sets are natural and diverse, and close to the instance similarity in the randomly sampled few-shot set. This indicates that our approach cannot be replaced by trivial implementations, such as aggregating a group of similar instances or picking out noise instances. Appendix G shows sampled instances from the searched few-shot set.

Table 3: The similarity analysis on the searched few-shot sets and randomly-sampled few-shot sets. For SSIM and MI, lower scores represent better diversity. For PSNR, higher scores represent better diversity. The searched few-shot sets show competitive diversity scores compared with randomly-sampled sets.

| Diversity Score | FFN-attack | VGG-attack | ResNet-attack | ResNeXt-attack | DenseNet-attack | Average-case |
|---|---|---|---|---|---|---|
| SSIM↓ | 0.06 | 0.12 | 0.18 | 0.20 | 0.12 | 0.12 |
| MI↓ | 0.44 | 0.44 | 0.37 | 0.35 | 0.46 | 0.44 |
| PSNR↑ | 54.57 | 57.25 | 56.18 | 58.04 | 56.03 | 57.43 |

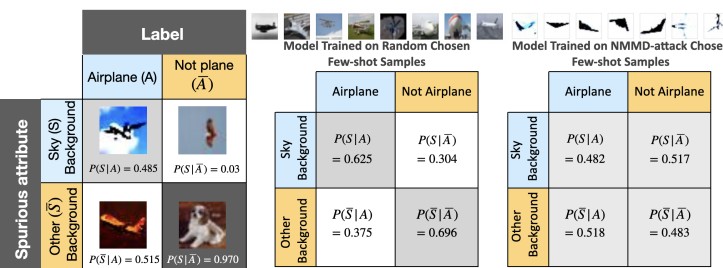

Figure 4: Case study of spurious correlation: Sky Background $\rightarrow$ airplane. Specifically, $P(S|A)$ refers to the number of airplane images with a sky background correctly judged, and so on. This spurious correlation is witnessed in the classification results of the model trained on a random-chosen few-shot set. The NMMD-attack results barely show this correlation.

## 7.2 Explaining the effectiveness of NMMD-attack with spurious correlation

Spurious correlation occurs as a statistical phenomenon, whereas confounders in data can be used to perform inference. These confounding attributes are often superficial features like background or lightning and are easy to capture. Here we compute the conditional probability of the spurious attribute to labels. All probabilities are statistically estimated by counting the figures with or without the hand-annotated spurious attribute. As shown in Figure. 4, airplanes in the CIFAR-10 dataset appear more common in a sky background than other classes, and thus models are likely to learn the spurious correlation "figure of airplanes $\rightarrow$ sky background $\rightarrow$ planes". These spurious correlations between training and test sets thus bring a high performance that is yet hard to generalize.

However, our worst-case few-shot set on CIFAR-10 picks several samples for each class as a few-shot set, all with an almost plain white background. This forbids models from learning to classify airplanes based on the sky background. The essential features are much harder to learn, resulting in much lower accuracy. Similar cases are common in various datasets. Other cases on Imagenet-1K include Bamboo Leaves $\rightarrow$ Small Panda, Water Background $\rightarrow$ Drake, and Human Activity $\rightarrow$ Paddle, as shown in Appendix F. All spurious correlation experiments are based on ResNeXt-attack set and finetuned with VGG model.

## 8 Conclusion

This paper proposes a worst-case evaluation to re-examine neural networks as few-shot learners by constructing a label-balanced subset from a full-size training set that results in the largest expected risks. An efficient method NMMD-attack is proposed to optimize the target in this work. Experiments show that NMMD-attack can find natural and diverse few-shot sets that successfully attack various architectures, even pre-trained models. The quantitative analysis gives several case studies to understand how spurious correlations between training and test sets affect few-shot evaluation. We find that the searched few-shot sets have fewer spurious attributes than randomly-sampled few-shot sets, which can explain why the searched few-shot set is more challenging. The worst-case evaluation re-examines the actual ability of neural networks on few-shot cases and also brings new problems to defense such attack for better robustness. Our work is still limited in the computer vision domain and we intend to apply to natural language tasks in the future. We also intend to further explore the negative impact of spurious attribute on general benchmarking and extend our theoretical analysis to the impact of distribution shift on spurious correlation.

## 9 REPRODUCIBILITY STATEMENT

- Experiment Reproducibility: We have included the code, data, and instructions to reproduce the experimental results. The codes and searched few-shot sets are available in the supplementary materials. We also record the gradient norms in the materials, which enables a quick re-implementation. Models and hyper-parameter settings are described in 6. For each result, we run experiments with different random seeds and report the average accuracy with standard deviation.

- Theory Reproducibility: The proofs of all propositions, lemmas, and theorems are included in A. A few theorems and lemmas referenced from other works are provided with careful citation.

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

# A PROOF

## A.1 PROOF FOR THEOREM 5.1

**Lemma 1.** *Define $\epsilon_S(h, f) := E_{x \sim S} |\delta(h(x)) - \delta(f(x))|$. For any hypothesis $h, h' \in \mathcal{H}$, there exists $\epsilon_H > 0$ which satisfies,*

$$|\epsilon_{P_{I_k}}(h, h') - \epsilon_P(h, h')| \leq MMD(\mathcal{H}, P_{I_k}, P) + \frac{\epsilon_{\mathcal{H}}}{2} \tag{11}$$

*$\epsilon_H$ is a constant for the complexity of hypothesis space.*

*Proof.*

$$
\begin{aligned}
\left| \epsilon_{P_{I_k}}(h, h') - \epsilon_P(h, h') \right| &\leq \sup_{h, h' \in \mathcal{H}} \left| \epsilon_{P_{I_k}}(h, h') - \epsilon_P(h, h') \right| \\
&= \sup_{h, h' \in \mathcal{H}} \left| \mathbf{P}_{\boldsymbol{x} \sim P_{I_k}} [\delta(h(\boldsymbol{x})) \neq \delta(h'(\boldsymbol{x}))] - \mathbf{P}_{\boldsymbol{x} \sim P} [\delta(h(\boldsymbol{x})) \neq \delta(h'(\boldsymbol{x}))] \right| \\
&= \sup_{h, h' \in \mathcal{H}} \left| \mathbf{P}_{\boldsymbol{x} \sim P_{I_k}} [h(\boldsymbol{x}) \neq h'(\boldsymbol{x})] - \mathbf{P}_{\boldsymbol{x} \sim P} [h(\boldsymbol{x}) \neq h'(\boldsymbol{x})] \right| \\
&= \sup_{h, h' \in \mathcal{H}} \left| \int_{\mathcal{X}} \mathbf{1}_{h(\boldsymbol{x}) \neq h'(\boldsymbol{x})} d\mu_{P_{I_k}} - \int_{\mathcal{X}} \mathbf{1}_{h(\boldsymbol{x}) \neq h'(\boldsymbol{x})} d\mu_P \right|
\end{aligned}
\tag{12}
$$

Inspired by proof in Peng et al. (2019), we use a continuous function to approximate the indicator function $\mathbf{1}_{h(\boldsymbol{x}) \neq h'(\boldsymbol{x})}$. Formally, for any $h, h'$, $\mathbf{1}_{h(\boldsymbol{x}) \neq h'(\boldsymbol{x})}$ is a $L^1$ function, thus there exists $f \in C_c(\mathcal{X})$ that satisfies,

$$
\begin{aligned}
\sup_{h, h' \in \mathcal{H}} \left| \int_{\mathcal{X}} \mathbf{1}_{h(\boldsymbol{x}) \neq h'(\boldsymbol{x})} d\mu_{P_{I_k}} - \int_{\mathcal{X}} \mathbf{1}_{h(\boldsymbol{x}) \neq h'(\boldsymbol{x})} d\mu_P \right| &\leq \left| \int_{\mathcal{X}} f(\boldsymbol{x}) d\mu_{P_{I_k}} - \int_{\mathcal{X}} f(\boldsymbol{x}) d\mu_P \right| + \frac{\epsilon_{\mathcal{H}}}{2} \\
&\leq \left| \sup_{f \in H} (E_{x \sim P}[f(x)] - E_{y \sim P_{I_k}}[f(y)]) \right| + \frac{\epsilon_{\mathcal{H}}}{2} \\
&= |\mathrm{MMD}(\mathcal{H}, P_{I_k}, P)| + \frac{\epsilon_{\mathcal{H}}}{2}
\end{aligned}
\tag{13}
$$

where $C_c(\mathcal{X})$ is a subset of the hypothesis space with compact support and $\epsilon$ is a constant for the complexity of hypothesis space which measures the closeness of the approximation.

**Lemma 2.** *Let $f_{I_k}$ be the trained classifier on the few-shot distribution $P_{I_k}$, and $f$ be the trained classifier on distribution $P$. Since $P_{I_k}$ is formed by a subset of the training examples, when training error $\epsilon_P(f) \to 0$ and $\epsilon_{P_{I_k}}(f_{I_k}) \to 0$, $\epsilon_{P_{I_k}}(f_{I_k}, f) \leq \epsilon_\alpha$, where $\epsilon_\alpha$ is a constant approaching zero.*

Following Ben-David et al. (2010), we use Lemma 1 and 2 to prove Theorem 5.1.

*Proof*

$$
\begin{aligned}
\epsilon_Q(f_{I_k}) &\leq \epsilon_Q(f) + \epsilon_Q(f_{I_k}, f) \\
&= \epsilon_Q(f) + \epsilon_{P_{I_k}}(f_{I_k}, f) + (\epsilon_Q(f_{I_k}, f) - \epsilon_P(f_{I_k}, f)) + (\epsilon_P(f_{I_k}, f) - \epsilon_{P_{I_k}}(f_{I_k}, f)) \\
&\leq \epsilon_Q(f) + \epsilon_{P_{I_k}}(f_{I_k}, f) + |\epsilon_P(f_{I_k}, f) - \epsilon_Q(f_{I_k}, f)| + |\epsilon_{P_{I_k}}(f_{I_k}, f) - \epsilon_P(f_{I_k}, f)| \\
&\leq \epsilon_Q(f) + \epsilon_\alpha + |\epsilon_P(f_{I_k}, f) - \epsilon_Q(f_{I_k}, f)| + \mathrm{MMD}(P_{I_k}, P) + \epsilon_{\mathcal{H}}
\end{aligned}
\tag{14}
$$

In which,

$$|\epsilon_P(f_{I_k}, f) - \epsilon_Q(f_{I_k}, f)| = \left| \int_{\mathcal{X}} \mathbf{1}_{f_{I_k}(\boldsymbol{x}) \neq f(\boldsymbol{x})} d\mu_P - \int_{\mathcal{X}} \mathbf{1}_{f_{I_k}(\boldsymbol{x}) \neq f(\boldsymbol{x})} d\mu_Q \right|$$

$$= |\sum_{i=1}^{n} \mathbf{1}_{f_{I_k}(\boldsymbol{x_i}) \neq f(\boldsymbol{x_i})} - E_Q \mathbf{1}_{f_{I_k}(\boldsymbol{x}) \neq f(\boldsymbol{x})}| \tag{15}$$

Here suppose test set $Q$ matches the distribution of data for this classification task, and $P$ is constructed by sampling $n$ i.i.d. samples from the distribution $Q$. Using Hoeffding inequality we have,

$$P(|\epsilon_P(f_{I_k}, f) - \epsilon_Q(f_{I_k}, f)| > t) \leq 2e^{-2nt^2} \tag{16}$$

Therefore, with a probability over $1 - 2e^{-2n\epsilon_t^2}$ (in practice $n$ is very large and the probability approaches 1), we have

$$\epsilon_Q(f_{I_k}) \leq \epsilon_Q(f) + \text{MMD}(P_{I_k}, P) + \epsilon_\alpha + t + \epsilon_{\mathcal{H}} \tag{17}$$

$\square$

## A.2    PROPERTY OF NTK

**Proposition A.1.** *(Neural Tangent Kernel, Arora et al. (2019)) Consider minimizing the loss $l(\theta)$ by gradient descent with infinitesimally small learning rate: $\frac{d\theta(t)}{dt} = -\nabla l(\theta(t))$. Let $u(t) = (f(\theta(t), x_i))_{i \in [n]} \in \mathbb{R}^{nd}$ be the network outputs on all $x_i$ at time $t$, and $Y = (y_i)_{i \in [n]}$ be the ground truth outputs, loss is $l(f(\theta, X), Y)$. Then $u(t)$ follows the following evolution, where $K^t$ is the NTK matrix, $K^t_{(i,j)} = \langle \frac{\partial f(\theta(t), x_i)}{\partial \theta}, \frac{\partial f(\theta(t), x_j)}{\partial \theta} \rangle$:*

$$u(t) - u(t-1) = K^t \frac{\partial l(\theta(t))}{\partial u(t)} + O((\theta_t - \theta_{t-1})^2) \tag{18}$$

The proof follows Arora et al. (2019).

## A.3    ON PROPOSITION 5.1

This proposition is adopted from Lemma.4 in Gretton et al. (2012).

**Lemma 3.** *(Restatement of Lemma.4 in Gretton et al. (2012)) Let $\mathcal{H}$ be the Reproducing Kernel Hilbert Space for kernel $K(\cdot, \cdot)$. If $K(\cdot, \cdot)$ is measurable and $E_{x \sim P} K(x, x) > 0, \forall x \in \mathcal{X}$, $E_{x \sim Q} K(x, x) > 0, \forall x \in \mathcal{X}'$, then mean embeddings $\mu_P$, $\mu_Q$ exists such that $E_x f = \langle f, \mu_P \rangle$ and satisfies,*

$$MMD(P, Q) = ||\mu_P - \mu_Q||_{\mathcal{H}} \tag{19}$$

The NTK kernel is a positive definite matrix and satisfies constraints in this Lemma. Also based on the definition of distance in RKHS, we can rewrite Equation. 19 into the form in Proposition. 5.1.

## A.4    PROOF FOR THEOREM 5.2

*Proof*

Using the MMD expression in integral form (Cheng & Xie, 2021), we can simplify the expression in Definition 5.1:

$$\text{MMD}_f(P, Q) = \sup_\theta \int_{\mathcal{X}} f(\theta, x) |\hat{p} - \hat{q}|(x) dx \tag{20}$$

where $\hat{p} = \frac{1}{n}\delta_{x_i}$, $x_i \sim P$, and $\hat{q} = \frac{1}{m}\delta_{y_i}$, $y_i \sim Q$. $\delta_x$ is the indicator function that equals to 1 when input equals to $x$, otherwise the output is 0.

Similarly we can write NTK-MMD initialized with $f(0)$ as,

$$\text{MMD}^2_{K_0}(P, Q) = \int_{\mathcal{X}} \int_{\mathcal{X}} K_0(x, x')(\hat{p} - \hat{q})(x)(\hat{p} - \hat{q})(x') dx dx' \tag{21}$$

We can see that $\text{MMD}_f$ involves solving an optimization problem. Here we design a loss to achieves the maximum at the last time step during fine-tuning. At the same time, we require the model $f$ to learn the distribution of the task.

$$L(f(\theta_t, x)) = \underbrace{-|\mathbb{E}_Q f(\theta(t), x) - \mathbb{E}_P f(\theta(t), x)|}_{\text{Unalignment loss}} + \underbrace{\mathbb{E}_P |f(x) - p(y|x)|}_{\text{Classification loss}} \tag{22}$$

During training, we take a two steps approach, first aligning the distributions and then minimizing the classification error (in practice, this can be done by training the representation encoder and then the classification head). Here we suppose at initialization, the loss is quite large such that the two distributions are aligned.

The NTK at time step $t$ is $K_t(x, x') = \langle \frac{\partial f(x, \theta(t))}{\theta(t)} \frac{\partial f(x', \theta(t))}{\theta(t)} \rangle$. Our $P$ is the training set distribution, and $P_{I_k}$ is the distribution $Q$ (here we write $Q$ for convenience). Using the L1 loss, combined with Proposition 3.1, we have:

$$u(x, t) - u(x, t - 1) = \nabla_{\theta_t} f(x; \theta_t)^T (\theta_t - \theta_{t-1}) + O\left(\|\theta_t - \theta_{t-1}\|^2\right)$$
$$= \int_{\mathcal{X}} \nabla_{\theta_t} f(x; \theta_t)^T \nabla_{\theta_t} f(x'; \theta_t)(-|\hat{p} - \hat{q}| + \hat{p})(x')dx' + O\left(\|\theta_t - \theta_{t-1}\|^2\right) \tag{23}$$

Recalling that, the NTK matrix remains stable during training, thus we can write it as,

$$u(x, t) - u(x, t - 1) = \eta \int_{\mathcal{X}} \nabla_{\theta_0} f(x; \theta_0)^T \nabla_{\theta_0} f(x'; \theta_0)(-|\hat{p} - \hat{q}| + \hat{p})(x')dx' + O\left(\|\theta_t - \theta_{t-1}\|^2\right) \tag{24}$$

Here $\mathcal{X}$ is the support set of $P$ and $Q$. Also,

$$\|\theta_t - \theta_{t-1}\| = \eta \|\nabla_{\theta_t} L(f(\theta(t)), x\|$$
$$= \frac{\eta}{n} \|\nabla_{\theta_t} f(\theta(t))\| \leq \frac{\eta L_f}{n} \tag{25}$$

Therefore, combining $t = cn$ steps (c epochs) of training we have:

$$u(x, t) - u(x, 0) = \eta \int_t \int_{\mathcal{X}} \nabla_{\theta_0} f(x; \theta_0)^T \nabla_{\theta_0} f(x'; \theta_0)(\hat{p})(x')dx' dt + \sum_{s=0}^{t} O\left(\frac{\eta^2 L_f^2 s}{n^2}\right)$$
$$= \eta \int_t \int_{\mathcal{X}} \nabla_{\theta_0} f(x; \theta_0)^T \nabla_{\theta_0} f(x'; \theta_0)(\hat{p})(x')dx' dt + O\left(\eta^2 L_f^2\right) + O(\eta^2 L_f^2) \tag{26}$$

Then combined with the definition of MMD, if $n \geq 2k$:

$$\text{MMD}_f(P, Q) = \int_{\mathcal{X}} u(x, t)|\hat{p} - \hat{q}|(x)dx$$
$$= \int_{\mathcal{X}} [u(x, 0) + \eta \int_t \int_{\mathcal{X}} \int_{\mathcal{X}} K^0(x, x')(-|\hat{p} - \hat{q}| + \hat{p})(x')dx'dt]|\hat{p} - \hat{q}|(x)dx + O(t\eta^2 L_f^2)$$
$$\leq 2\eta \int_t \int_{\mathcal{X}} \int_{\mathcal{X}} K^0(x, x')|\hat{p} - \hat{q}|(x')dx'dt|\hat{p} - \hat{q}|(x)dx + O(t\eta^2 L_f^2)$$
$$= 2\eta t \text{MMD}_{K^0}^2(P, Q) + O(t\eta^2 L_f^2) \tag{27}$$

Also,

$$
\begin{aligned}
\text{MMD}_f(P, Q) &= \int_{\mathcal{X}} u(x, t)|\hat{p} - \hat{q}|(x)dx \\
&= \int_{\mathcal{X}} [u(x, 0) + \eta \int_t \int_{\mathcal{X}} \int_{\mathcal{X}} K^0(x, x')(-|\hat{p} - \hat{q}| + \hat{p})(x')dx'dt]|\hat{p} - \hat{q}|(x)dx + O(t\eta^2 L_f^2) \\
&> \eta \int_t \int_{\mathcal{X}} \int_{\mathcal{X}} K^0(x, x')|\hat{p} - \hat{q}|(x')dx'dt|\hat{p} - \hat{q}|(x)dx + O(t\eta^2 L_f^2) \\
&= \eta t \text{MMD}_{K^0}^2(P, Q) + O(t\eta^2 L_f^2)
\end{aligned}
\tag{28}
$$

$\square$

Table 4: Hyperparameters settings. These settings are reported in their official repository for *best practice*. CA refers to cosine annealing scheduler  (Loshchilov & Hutter, 2016). The Linear refers LinearLR scheduler in Pytorch. OneCycle refers 1-cycle learning rate policy (Smith & Topin, 2019).

| Models | Datasets | Batch Size | Training Epochs | Optimizer | Learning Rate |
|---|---|---|---|---|---|
| FFN | MNIST | 128 | 50/200 | SGD | 0.01  (CA) |
| | CIFAR-10 | 128 | 50/200 | SGD | 0.01 (CA) |
| | CIFAR-100 | 128 | 50/200 | SGD | 0.01 (CA) |
| | ImageNet-1K | 32 | 40/90 | SGD | [0.01, 0.001, 0.0001] |
| VGG | MNIST | 128 | 50/200 | SGD | 0.01  (CA) |
| | CIFAR-10 | 128 | 50/200 | SGD | 0.01 (CA) |
| | CIFAR-100 | 128 | 50/200 | SGD | 0.01 (CA) |
| | ImageNet-1K | 32 | 40/90 | SGD | [0.01, 0.001, 0.0001] |
| ResNet | MNIST | 128 | 50/200 | SGD | 0.01  (CA) |
| | CIFAR-10 | 128 | 50/200 | SGD | 0.01 (CA) |
| | CIFAR-100 | 128 | 50/200 | SGD | 0.01 (CA) |
| | ImageNet-1K | 32 | 40/90 | SGD | [0.1, 0.01, 0.001] |
| ResNeXt | MNIST | 128 | 50/200 | SGD | 0.01  (CA) |
| | CIFAR-10 | 128 | 50/200 | SGD | 0.01 (CA) |
| | CIFAR-100 | 128 | 50/200 | SGD | 0.01 (CA) |
| | ImageNet-1K | 100 | 40/90 | SGD | [0.1, 0.01, 0.001] |
| DenseNet | MNIST | 128 | 50/200 | SGD | 0.01  (CA) |
| | CIFAR-10 | 128 | 50/200 | SGD | 0.01 (CA) |
| | CIFAR-100 | 128 | 50/200 | SGD | 0.01 (CA) |
| | ImageNet-1K | 32 | 40/90 | SGD | [0.1, 0.01, 0.001] |
| ViT-B/16 | CIFAR-10 | 32 | 4 | Adam | 5e-5 (Linear) |
| | CIFAR-100 | 32 | 4 | Adam | 5e-5 (Linear) |
| EfficientNetV2-S | CIFAR-10 | 32 | 4 | AdamW | 1e-3 (OneCycle) |
| | CIFAR-100 | 32 | 4 | AdamW | 1e-3 (OneCycle) |

## B  Models and Hyperparameters

To evaluate the robustness of over-parameterized neural networks, we consider the following models. 1) **FFN**, a feed-forward neural network with two convolution and pooling layers and three feed-forward layers. 2) **VGG** (Simonyan & Zisserman, 2014), a classical convolutional neural network. We use the VGG-16 with 13 convolution layers and three fully connected layers as implementation. 3) **ResNet** (He et al., 2016), a residual neural network. We use the ResNet-18 with 16 residual blocks, one convolution layer, and one fully connected layer as implementation; 4) **ResNeXt** (Xie et al., 2017) incorporating the advantages of ResNet and Inception (Szegedy et al., 2015; 2016; 2017; Ioffe & Szegedy, 2015). We use the ResNeXt-29 (2x64d) for MNIST, CIFAR-10, and CIFAR-100, and ResNeXt-50 (32x4d) for ImageNet. 5) **DenseNet** (Huang et al., 2017). We use DenseNet-121 with 121 layers, one convolution layer, and one fully connected layer as re-implementation.[1] Besides, to verify the attack ability NMMD-attack on the pre-trained models, we also re-implement two pre-trained models: 1) Transformer-based **ViT** (Dosovitskiy et al., 2021)[2] and 2) Convolutional-based **EfficientNetV2** (Tan & Le, 2021)[3]. For FFN, VGG, ResNet, ResNeXt, and DenseNet on ImageNet, we resize all the images into $256 \times 256$ and then center-crop them into $224 \times 224$. For ViT on CIFAR, we resize all the images into $224 \times 224$, while $384 \times 384$ for EfficientNetV2.

We list hyper-paramter settings in Table 4. All the SGD optimizers are with a momentum of 0.9. For Adam/AdamW, we set $\beta = (0.9, 0.999)$. For the learning rate in selected Imagenet, the milestones are [15, 30], while [30, 60] for the full-sized. We conduct all the experiments on a single A100 GPU.

---

[1]For VGG, ResNet, ResNeXt, and DenseNet on CIFAR and MNIST, we use the implementation from `https://github.com/kuangliu/pytorch-cifar`. As for ImageNet, we use the implementation from torch.models.

[2]We use the implementation from `https://huggingface.co/google/vit-large-patch16-224`

[3]We use the implementation from torch.models.

## C    DETAILS OF SIMILARITY MEASURE METHODS

1) Structural SIMilarity (SSIM), which is a widely recognized method for measuring the similarity between two images (Wang et al., 2004). The SSIM index considers three measurements on the two images $x$ and $y$, including luminance, contrast, and structure. Following the settings of Wang et al. (2004), we formulate SSIM by:

$$\mathcal{SSIM}(x, y) = l^{\alpha} \cdot c^{\beta} \cdot s^{\gamma} \tag{29}$$

We set the weights $\alpha$, $\beta$, and $\gamma$ all to 1.

2) Mutual Information (MI), which is a measure of the mutual dependence between the two variables in information theory. Following (Studholme et al., 1999), we calculate MI for image matching. Given the signal intensity in an image, it evaluates how well the signal in the other image will be predicted. Compared with other measurements, MI has fewer restrictions on image modality and alignment. We implement MI by applying the normalized mutual information function in scikit-learn [4] toolkit and normalizing MI to the interval 0 to 1.

3) Peak Signal-to-Noise Ratio (PSNR) is a frequently used metric for image quality comparison between two images, especially in the area of image compression. PSNR computes the mean-square error of the compressed and the original image and further calculates the peak error by:

$$\mathcal{PSNR} = 10 \log_{10} \left( \frac{R^2}{MSE(x, y)} \right) \tag{30}$$

where $R$ is the maximum fluctuation in the image data type. In our experiment, we normalized the images, took the MSE of the images in the worst-case group one-to-one, and set the $R$ to 1.

---

[4] https://scikit-learn.org

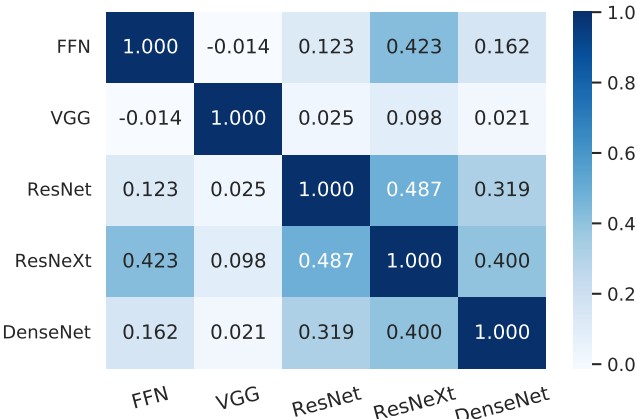

Figure 5: Spearman correlation of gradient norms calculated by different models. All results have $r < 0.005$ except for the VGG correlation, whose $r > 0.05$.

## D  INTER-MODEL CORRELATION ANALYSIS

For different models, the gradient norms are different and thus various models can generate different adversarial few-shot sets.In this section, we explore whether there is an intrinsic gradient ranking correlation across different models.

We report the correlation analysis between different models in Figure 6. As can be seen from the graph, the VGG is not correlating with any of the other models' rankings. The correlation result also aligns with our main results that the VGG attack is almost invalid. Beyond that, we can see that ResNet, DenseNet, and ResNeXt, which have similar residual structures, maintain some correlation. Apart from that, the FFN, which also has a strong attack capability, only maintains a high correlation with ResNeXt, showing that influential ranking is model-independent.

### D.1  WORST-CASE EVALUATION ON FEW-SHOT LEARNING MODELS

We also explored the ability of the NMMD-Attack on the FEW-SHOT learning models. We selected the ProtoNet (Snell et al., 2017)[5] for experiments on CIFAR-100 with FFN-attack. From the results, we can see that even though the model is designed for few-shot learning, it still suffers from our attack.

Table 5: The attack results on few-shot models on CIFAR-100. The abbreviations follow Table 1.

| Models | Average-case | | NMMD-attack | | Test Acc. Gap | |
|---|---|---|---|---|---|---|
| | Train Acc. | Test Acc. | Train Acc. | Test Acc. | Abs. | Rel. |
| ProtoNet | 99.97 | 54.88 | 99.96 | **52.46** | 2.42 | 4.41 |

---

[5]We use the implementation from `https://github.com/orobix/Prototypical-Networks-for-Few-shot-Learning-PyTorch`

Table 6: The comparison between average performance and attack performance of different size on CIFAR-10. The abbreviations follow Table 1.

| k-shot | Models | Average-case | | NMMD-attack | | Test Acc. Gap | |
|---|---|---|---|---|---|---|---|
| | | Train Acc. | Test Acc. | Train Acc. | Test Acc. | abs. | rel. |
| 50-shot | FFN | 52.16 ±10.93 | 30.13 ±1.60 | 84.52 ±9.82 | **24.90** ±1.00 | **5.23** | **17.36** |
| | VGG-16 | 100.00 ±0.00 | 40.00 ±0.41 | 100.00 ±0.00 | **22.14** ±2.02 | **17.89** | **44.65** |
| | ResNet-18 | 100.00 ±0.00 | 35.06 ±0.46 | 100.00 ±0.00 | **19.01** ±1.17 | **10.83** | **45.78** |
| | ResNeXt-29 | 100.00 ±0.00 | 34.95 ±1.15 | 100.00 ±0.00 | **24.12** ±0.59 | **10.83** | **30.99** |
| | DenseNet-121 | 100.00 ±0.00 | 40.57 ±1.75 | 100.00 ±0.00 | **24.29** ±1.88 | **16.28** | **40.13** |
| 200-shot | FFN | 100.00 ±0.00 | 43.08 ±1.29 | 99.99 ±0.01 | **36.12** ±1.68 | **6.06** | **16.16** |
| | VGG-16 | 100.00 ±0.00 | 56.93 ±0.41 | 100.00 ±0.00 | **34.93** ±1.24 | **22.00** | **38.64** |
| | ResNet-18 | 100.00 ±0.00 | 53.65 ±0.88 | 100.00 ±0.00 | **34.42** ±1.69 | **19.23** | **35.84** |
| | ResNeXt-29 | 100.00 ±0.00 | 52.07 ±1.10 | 100.00 ±0.00 | **34.66** ±1.19 | **17.41** | **33.44** |
| | DenseNet-121 | 100.00 ±0.00 | 60.85 ±0.82 | 100.00 ±0.00 | **36.18** ±1.42 | **24.67** | **40.54** |
| 2000-shot | FFN | 100.00 ±0.00 | 59.10 ±0.69 | 99.99 ±0.01 | **55.84** ±0.90 | **3.26** | **5.52** |
| | VGG-16 | 100.00 ±0.00 | 82.66 ±0.15 | 100.00 ±0.00 | **78.22** ±0.29 | **4.44** | **5.37** |
| | ResNet-18 | 100.00 ±0.00 | 80.68 ±0.25 | 100.00 ±0.00 | **76.49** ±0.77 | **4.19** | **5.19** |
| | ResNeXt-29 | 100.00 ±0.00 | 79.10 ±0.35 | 100.00 ±0.00 | **76.00** ±0.37 | **3.10** | **3.92** |
| | DenseNet-121 | 100.00 ±0.00 | 85.96 ±0.45 | 100.00 ±0.00 | **82.64** ±0.34 | **3.32** | **3.86** |

# E   ATTACK ON THE FEW-SHOT SETS WITH DIFFERENT SIZES

We conducted attack experiments on few-shot sets with different sizes. The results show that with the increase of training data, the robustness gradually becomes better. There is considerable overlap between the attack and random datasets at larger data sizes. The performance drop on 2000-shot cases also can indicate the effectiveness of our methods.

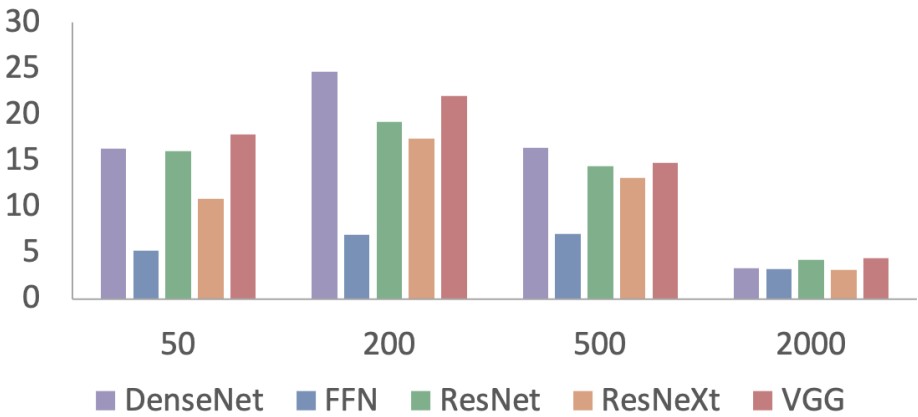

Figure 6: Attack performances on different scale subsets of CIFAR-10. The bars represent the gap between NMMD-attack and average-case.

# F    Spurious Correlation Analysis

We case study the correlation between several spurious attributes and labels on the Imagenet-1K. The results in Figures 6,7&8 shows that spurious correlations are common. When trained on a randomly selected few-shot set, all three cases suffer from spurious correlations at different levels. However, the spurious correlations problem becomes less severe after training on our NMMD-attack set in Bamboo Leaves VS. Small Panda and Water Background VS. Drake cases. Compared to results on CIFAR-10, the spurious correlation alleviation is less precise, possibly because the model has not learned the spurious attributes on Imagenet well. Furthermore, the spurious attributes on complicated figures may also be more complicated and hard to gain.

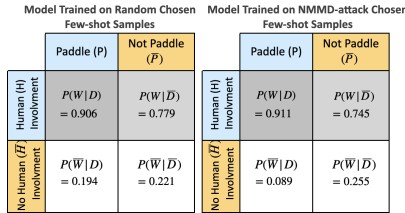

Figure 7: Case study of spurious correlation: Human Involvement → Paddle.

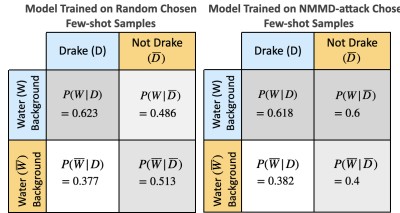

Figure 8: Case study of spurious correlation: Water Background → Drake.

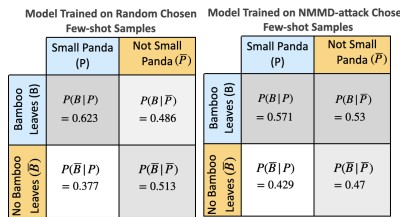

Figure 9: Case study of spurious correlation: Bamboo Leaves → Small Panda.

# G VISUALIZATION OF THE FEW-SHOT SET SEARCHED BY NMMD-ATTACK

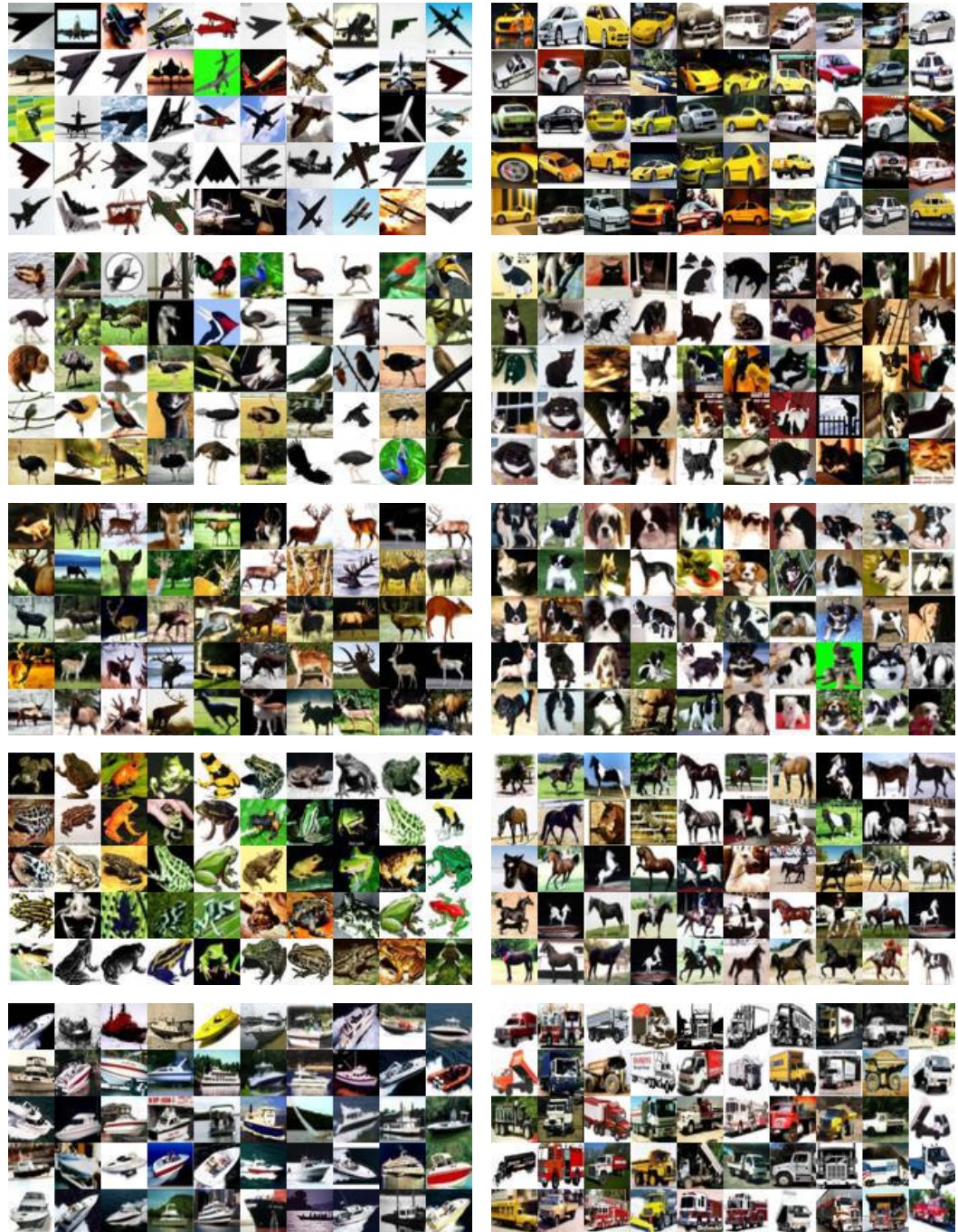

Figure 10: Visualization of the few-shot set searched by NMMD-attack for CIFAR-10. We randomly choose 50 examples for each label.

