# OpenReview forum: "Worst-case Few-shot Evaluation: Are Neural Networks Robust Few-shot Learners?"
_ICLR.cc/2023/Conference — Submitted to ICLR 2023_

### Official Review · Reviewer_Hiwt · 2022-10-14

**Confidence:** 4
**Correctness:** 3
**Technical Novelty And Significance:** 2
**Empirical Novelty And Significance:** 1
**Recommendation:** 1

**Clarity, Quality, Novelty And Reproducibility:**

Clarity: The paper is relatively clear.
Reproducibility: The paper is reproducible to the extent I see.
Quality: I am not sure what is meant by quality. If this is an overall rating of the work, then I think this work does not meet the bar for ICLR.
Novelty: The idea of maximizing MMD is novel, but none of the experimental findings are novel.

**Strength And Weaknesses:**

Strengths:
1. The problem of finding worst case performance is an important one. There is not much analysis of how training sets impact downstream accuracy, so this is a step in the right direction.
2. The idea of using NTK to get around the intractability of MMD is clever, and leads to a practical algorithm.

Weaknesses:
1. This paper positions itself as few-shot evaluation. Yet, experiments are with 50-500 examples per class, which is actually a moderately large training set. This is not few-shot evaluation in any shape or form.
2. In addition, while the paper talks of evaluating few-shot models, *not even a single few-shot learning system is evaluated*. Instead, the paper chooses to evaluate standard classification approaches that are explicitly designed to work with large training sets. As such, the fact that these networks lose performance in the worst case evaluation is unsurprising and yields no useful insight. Few-shot learning techniques are intended to work well with small training sets, and may in fact be more resilient in this worst case evaluation. Comparing different few-shot learning techniques would yield some insight re:possible benefits of few-shot learning.
3. At the very least, with standard classification approaches, there are knobs such as data augmentation and weight decay that one can change. There are also ways of training the network with adversarial data augmentation. I would expect at least some analysis along these lines to get some insight into what affects worst case performance. No such analysis is present.
4. The fact that the worst case performance is going to be low is obvious just from the average case numbers: with the exception of MNIST, all models are clearly overfitting dramatically on the training set (see large gap between average case test and train performance in Table 1). It is therefore not surprising at all that the test-train gap will be even larger when the training set is adversarially chosen. This further underscores the need to evaluate regularization/augmentation.
5. The one novel contribution of this paper is an approach to produce adversarially sampled training sets. But this is not evaluated well. The only baseline presented is average case performance. Here is a simple baseline that I would like to see: sample N different training sets, and report the *minimum* accuracy over all N.
6. In practical scenarios, one usually has some control over what images to get labeled. I was hoping for some insight on this, and some general advice to downstream applications on how to intelligently sample a good training set. This is once again missing.



**Summary Of The Paper:**

This paper aims to evaluate the robustness of recognition systems to adversarially sampled training sets, and thus get an estimate of the worst case performance. The main contribution is an approach to identify the worst possible training set; this approach is based on maximizing the MMD between the training set and the test set, and introduces approximations based on the neural tangent kernel for tractably working with the MMD. Results are shown on four standard datasets and a few different model architectures.

**Summary Of The Review:**

This paper is a step towards an important problem: understanding the robustness of neural networks to different training sets. However, in its current form it offers no useful insights and has minimal experiments that are not relevant to the stated title or thesis. Therefore it does not meet the bar.

The issues in this paper are too significant to be addressed in the rebuttal. I recommend that the authors address the weaknesses above and resubmit to a future conference.

---

> ### Author Response · Authors · 2022-11-07
> **For the sake of review quality, we expect that the reviewer can reconsider so many claims made by wrong evidence**
>
> **We are always happy to receive sharp comments. These suggestions can help us make the paper better.  However, for the sake of review quality, we also expect that the reviewer can draw conclusions based on right evidence.**
>
>
> [1] This paper positions itself as few-shot evaluation. Yet, experiments are with 50-500 examples per class, which is actually a moderately large training set. This is not few-shot evaluation in any shape or form.
>
> Following recent work  [1] [2] [3] [4], we put our work under few-shot settings. We know that traditional few-shot models are usually based on 10-shot or fewer. Recently, few-shot learning has been extensively explored. The number of shots is increased to 32, 50, and more [1] [2] [3] [4]. Since there is a trend to fine-tune a model on limited data, we mainly focus on this line to evaluate their worst-case performance. We conduct experiments on larger-size data to show how the training data size affects worst-case performance. Also, we conduct experiments over a traditional few-shot framework, ProtoNet. The reviewer can see the results in Table 5 in Appendix D.
>
>
> 1. Flamingo: a Visual Language Model for Few-Shot Learning. k=32.  DeepMind 2022. Citation: 93.
> 2. GPT-3: Language Models are Few-Shot Learners. k=50.
> 3. It’s Not Just Size That Matters: Small Language Models Are Also Few-Shot Learners. K=32.
> 4. FLEURS: Few-shot Learning Evaluation of Universal Representations of Speech. 12 hours of speech supervision per language.
>
>
>
> [2] not even a single few-shot learning system is evaluated.
>
> **It is a severe but false judgment**. **The reviewer missed our experiments on a single few-shot learning system**. We conduct experiments over a widely-used few-shot learning system, ProtoNet. The reviewer can see the results in Table 5 in Appendix D.  In the next version, we will add more experiments.
>
> [3]  As such, the fact that these networks lose performance in the worst-case evaluation is unsurprising and yields no useful insight.
>
> **It is a true judgment but has nothing to do with our contribution**. It is true that worst-case evaluation loses performance. Our work is not to prove worst-case performance is worse than average-case performance, we try to find a solution to estimate the worst-case performance instead.
>
> [4] I would expect at least some analysis along these lines to get some insight into what affects worst-case performance. No such analysis is present.
>
> **It is a severe but false judgment**. **We analyzed how widely-used robust solutions affect worst-case performance: increasing data / pre-trained models**. These results are shown in Table 2 and Table 6.
>
> [5] The fact that the worst case performance is going to be low is obvious just from the average case numbers: with the exception of MNIST, all models are clearly overfitting dramatically.
>
> **We will appreciate it if the reviewer could carefully read our motivation and contribution.**
>
> Previous work usually explores reducing the gap between test performance and training performance. It represents how good a model can be. However, such promising performance is over-estimated due to spurious correlations.
>
> We focus on the worst-case to see how bad a model can be without the effects of spurious correlations.
>
> These are two different things.
>
> There exists a dangerous case that current average performance does not evaluate the over-fitting problem well, Like MNIST and CIFAR10. MNIST is a good case showing how spurious correlation affects our judgment about a model: the large gap between average performance and worst-case performance.
>
>
> [6] The one novel contribution of this paper is an approach to produce adversarially sampled training sets. But this is not evaluated well.
>
> **It is a severe but false judgment**.   **Actually, we conducted this baseline. Average case performance is the simple baseline you mentioned.** We uniformly sample m sets from training data and report the average performance and variance over $m$ sets in Table 1.
>
>
> [7] In practical scenarios, one usually has some control over what images to get labeled. I was hoping for some insight on this, and some general advice to downstream applications on how to intelligently sample a good training set. This is once again missing.
>
> **Actually, we have shown evidence to see what is a good training set from a spurious correlation perspective**. Appendix F and Figure 4 show that spurious correction sometimes comes from "background" problems. We can consider diverse backgrounds when annotating data.
>
>
>
> [8] Novelty: none of the experimental findings are novel.
>
> We expect that the reviewer can reconsider the evaluation perspective about experiments. The experiments are used to show the effectiveness of the proposed approach. "not novel" is not an appropriate word to evaluate experiment quality.
>
>
> [9] Quality: I am not sure what is meant by quality.
>
> **Please read the reviewer guideline before reviewing a paper.**

---

> > ### Comment · Reviewer_Hiwt · 2022-11-07
> > **Response**
> >
> > Please see inline responses below.
> > [1] The work the authors  point to is in language modeling, but what the paper evaluates on is on image recognition. In 5 years of working on few-shot learning, I have found that unless there is a domain difference or severe overfitting, the 50 example regime is phenomenologically different from the 5 example regime. In image recognition, few-shot relates to <20 examples per class. I am ok with the paper looking at the 50-shot case, but any few-shot learning paper has to look at the <20-shot case as well, just because the regimes are so different.
> >
> > If the authors wish to evaluate using language models, please go ahead. But the kinds of problems that are considered in the two communities are very different. Comparing to the works cited requires evaluating language modeling-based techniques.
> >
> > [2] Prototypical networks are not sufficient  for few-shot learning, especially when the claim is about few-shot learning techniques in general. At the very least, I would expect evaluations of (a) models that explicitly alter feature representations based on the available training set, e.g., FEAT[A], and (b) models that explicitly encode translation invariance, e.g., FRN[B] or DeepEMD[C]. There are other models in the literature as well that have many different invariance properties; see the related work of more of these approaches.
> >
> > [4] When I mean analysis of what affects worst case performance, I specifically asked about data augmentation strategies or weight decay, or other regularization strategies. Regularization is fundamentally related to robustness. Any evaluation of worst-case performance should include an evaluation of the impact of regularization during training. Table 4 talks about pre-trained models and Table 6 talks about the number of shots. Neither looks at regularization.
> >
> > [5] "We uniformly sample m sets from training data and report the average performance and variance over m sets in Table 1."
> >
> > What I requested is:
> > "We uniformly sample m sets from training data and report the *minimum* performance and variance over m sets in Table 1."
> >
> > [7] I would expect this observation about correlations with background to be experimentally verified. Namely, on a new dataset (Not CIFAR), how would one choose training images? Would this new strategy of choosing training images lead to higher accuracy?
> >
> > In sum: my concerns in this paper are not addressed. I am still of the opinion that this paper needs to be substantially revised and resubmitted.
> >
> > Note that I do consider the goal of modeling and understanding worst case performance to be a noble and important one. However, I do not believe the paper in its current form does justice.
> >
> > References:
> > A: Ye, Han-Jia, et al. "Few-shot learning via embedding adaptation with set-to-set functions." Proceedings of the IEEE/CVF Conference on Computer Vision and Pattern Recognition. 2020.
> >
> > B: Wertheimer, Davis, Luming Tang, and Bharath Hariharan. "Few-shot classification with feature map reconstruction networks." Proceedings of the IEEE/CVF Conference on Computer Vision and Pattern Recognition. 2021.
> >
> > C: Zhang, Chi, et al. "Deepemd: Few-shot image classification with differentiable earth mover's distance and structured classifiers." Proceedings of the IEEE/CVF conference on computer vision and pattern recognition. 2020.

---

> > > ### Author Response · Authors · 2022-11-08
> > > **Is it appropriate to fully negate a paper just because it does not conduct optional experiments and does not conduct experiments out of paper scope?**
> > >
> > > [1] Few-shot settings
> > >
> > > We thank the reviewer for pointing out the difference between our paper and the traditional few-shot setting. It is a good suggestion.
> > >
> > > We focus on evaluating the learning ability of neural networks under limited data. It has been a hot research line evaluating the learning ability of a new model on limited data. However, current benchmarks usually have the problem of "false high performance". We aim to address this problem by providing a worst-case evaluation benchmark.
> > >
> > > **To avoid misunderstanding our work, as suggested by Reviewer #2 and Reviewer #3, we will change few-shot learning into learning with limited data.**
> > >
> > > [2] Prototypical networks are not sufficient for few-shot learning
> > >
> > > We are glad that the reviewer mentioned our results of prototypical networks.
> > >
> > > As explained in response [1],  we focus on evaluating neural networks under limited data (shot > 10) without assuming the maximum number of shots is 10 or less.
> > >
> > > Traditional few-shot learning limits their scope to 10-shot or less. By clarifying the concept difference, **these models are not our target settings, and more experiments on traditional few-shot networks become less important.**
> > >
> > >
> > >
> > > [3]  When I mean analysis of what affects worst case performance, I specifically asked about data augmentation strategies or weight decay, or other regularization strategies. Any evaluation of worst-case performance should include an evaluation of the impact of regularization during training.
> > >
> > > We take these suggestions and will do weight decay and more data augmentation strategies. But, still, we also would like to ask the reviewer question: **Is it appropriate to fully negate a paper just because it does not conduct an optional experiment?**
> > >
> > > **Our contribution: a worst-case benchmark and a solution to estimate the worst-case set.**
> > >
> > > What are necessary experiments: the comparison to show the effectiveness of the proposed solution. We have done.
> > >
> > > What are discussion experiments:  try robust improvement methods to evaluate whether these methods can improve worst-case performance. We also have done.
> > >
> > > The reviewer confuses two concepts: "what affects worst-case performance" and "regularization strategies". "Neither looks at regularization" does not mean we do not analyze "what affects worst-case performance".
> > >
> > > Do we need to analyze what affects worst-case performance? Yes.
> > >
> > > We analyze two widely used solutions: pre-training and augmenting more data. Nowadays,  pre-trained networks have been a solid backbone supporting all kinds of applications. They learn knowledge from massive data and have been proven to have better generalization. Therefore, we consider pre-trained networks to answer "what affects worst-case performance".
> > >
> > > **Do we need to analyze how weight decay and data augmentation affects worst-case performance? We believe that it is optional.**
> > >
> > > There are many other solutions that can improve robustness. We only have 9 pages. There are always uncovered methods. **If we have done A and B, is missing C really unacceptable?**
> > >
> > >
> > > [4] What I requested is: "We uniformly sample m sets from training data and report the minimum performance and variance over m sets in Table 1."
> > >
> > > **What we say is exactly what the reviewer requested**.  We **did this baseline** but reported mean and variance.  Generally speaking, the minimum performance can be estimated by mean and standard variance. Therefore, it is a common setting to report mean and variance.  As the reviewer required, we report the minimum performance in the following table.  The high performance shows that random sampling does not estimate the worst-case bound.  Please see the following table.
> > >
> > >
> > > |          | mnist | cifar10 | cifar100 | imagent |
> > > |----------|-------|---------|----------|---------|
> > > | ffn      | 97.37 | 48.34   | 13.69    | 4.68    |
> > > | vgg      | 98.48 | 64.84   | 26.38    | 13.90   |
> > > | resnet   | 98.55 | 61.10   | 24.63    | 29.71   |
> > > | resnext  | 98.24 | 60.95   | 23.35    | 37.87   |
> > > | densenet | 99.07 | 70.1    | 31.94    | 36.86   |
> > >
> > >
> > > [5] I would expect this observation about correlations with background to be experimentally verified.
> > >
> > > All previous studies have shown that spurious correlations are temporarily only obtainable through human knowledge. We are interested in verifying this, yet temporarily we can only focus on the worst-case evaluation part and leave this for future work.
> > >
> > > [6] Namely, on a new dataset (Not CIFAR), how would one choose training images? Would this new strategy of choosing training images lead to higher accuracy?
> > >
> > > **It is a good question but out of our paper scope**. We will explore it in follow-up works. Our paper focuses on the worst-case evaluation scope. It is not the necessary duty for the current version to give insights for questions that are not covered by the worst-case evaluation scope.

---

> > > > ### Comment · Reviewer_Hiwt · 2022-11-08
> > > > **Response**
> > > >
> > > > Thanks for engaging.
> > > >
> > > > A reframing of the paper would certainly help. In fact, I would remove even the phrasing of "limited data". My whole point is that for this to be a paper about "few shot" or "limited data", the paper should engage much more with methods people use in the literature for limited data. If the paper was framed as "worst case evaluation of neural networks", I wouldn't have as many concerns.
> > > >
> > > > As it stands however, I don't think the paper contributes enough to reach the ICLR bar. In the current framing, evaluation of other few shot learning techniques (and the many different kinds of data augmentations and regularizations peoplle use in practice) is *necessary*. If the framing is limited data, necessary experiments would also include semi or self supervised learning.
> > > >
> > > > An alternative framing that doesnt focus on limited data may resolve this issue, but is too big of a change to justify acceptance.
> > > >
> > > > Reframing and resubmitting is the best path forward in my opinion.

---

> > > > > ### Author Response · Authors · 2022-11-08
> > > > > **We respect the reviewer's right to give any decisions.  We also expect that the reviewer can draw reasonable conclusions based on reasonable evidence.**
> > > > >
> > > > > We thank the reviewer for the sharp comments. Actually, we appreciate all reviewers giving us valuable and constructive feedback.
> > > > >
> > > > > For us, it does not matter whether the paper is accepted.   In AI community, acceptance is not a hard thing. It is just a matter of time, and revison makes the paper better. Making high-quality paper is the hardest thing. We are happy to receive some sharp and constructive comments. These comments can help us make the paper better.
> > > > >
> > > > > For the reviewer, we believe that it is the reviewer's right to give a final decision based on any reasons.  We respect this right. We do not aim to convince the reviewer that they must buy our contributions and arguments.
> > > > >
> > > > > However, as authors, we also have the right to defend the paper.
> > > > >
> > > > > [1] In fact, I would remove even the phrasing of "limited data".
> > > > >
> > > > > Removing "limited data" is not our setting. We design the worst-case evaluation for evaluating learning ability under limited data.
> > > > >
> > > > >
> > > > > [2]  I don't think the paper contributes enough to reach the ICLR bar. In the current framing, evaluation of other few shot learning techniques (and the many different kinds of data augmentations and regularizations people use in practice) is necessary. If the framing is limited data, necessary experiments would also include semi or self supervised learning.
> > > > >
> > > > > The major controversial part lies in **whether the analysis of data augmentation, regularizations, semi or self-supervised learning, is necessary**.
> > > > >
> > > > > First,  as we wrote in the previous response, it is a good suggestion and we are conducting these experiments and will add these results to our paper.
> > > > >
> > > > > **Second, we argue that it is still optional.**
> > > > >
> > > > > Our paper is an evaluation paper by proposing a worst-case evaluation benchmark and a solution to find the worst-case set.
> > > > >
> > > > > We believe that the necessary experiments should be experiments showing the effectiveness of the proposed solution. We have done this. In addition, we also discuss the correlation analysis between different models, the quality of the search set, the spurious correlations in the search set, and the theoretical formation. All these experiments show the effectiveness of the proposed method and the quality of the search set.  **We are glad that at least the reviewer and the authors reached an agreement about the benchmark quality and the novelty of the method.**
> > > > >
> > > > > We believe that further discussion about current robustness tricks based on the proposed worst-case benchmark is optional.  **Here "optional" means that it should be encouraged to do. If the authors miss one of them, the paper should not be punished too much.** we indeed do some analysis, including pre-training and adding more data.
> > > > >
> > > > > Similar things happened in adversarial attack literature. Many previous robust evaluation studies are accepted only based on evaluation experiments without any analysis of defensive tricks, though there are a lot of defense approaches before the publishing time.  We take adversarial attack studies [1][2][3][4][5] as examples.  **If the reviewer rejected these studies because they did not conduct experiments on defense tricks, we would lose so many good papers.** By taking these studies as examples, we just want to stress the difference between necessary experiments and optional experiments.
> > > > >
> > > > > 1. Adversarial Attacks on Neural Network Policies. Citation: 640
> > > > > 2. Audio Adversarial Examples: Targeted Attacks on Speech-to-Text. Citation: 951.
> > > > > 3. Synthesizing Robust Adversarial Examples. Citation: 1280.
> > > > > 4. Adversarial Machine learning at scale. Citation: 2415.
> > > > > 5. Adversarial examples in the physical world. Citation: 4263.
> > > > >
> > > > >
> > > > >
> > > > > We have accepted all constructive suggestions and will revise the paper to make it better.  Also, again, we respect the reviewer's right to give any decisions.

---

> > > > > > ### Comment · Reviewer_Hiwt · 2022-11-08
> > > > > > **Thanks**
> > > > > >
> > > > > > Of course the authors have every right to defend the paper as well. I appreciate the time the authors have taken to engage with the criticism.
> > > > > >
> > > > > > I agree that the disagreement here boils down to what is necessary and what is optional. I have a particular bar for this question (specifically given my experience with few shot learning and learning from few labels) which informs my rating. Of course, the final decision is in the hands of the AC who may have their own expectations.
> > > > > >
> > > > > >
> > > > > > Thanks again.

---

### Official Review · Reviewer_di2U · 2022-10-23

**Confidence:** 3
**Correctness:** 2
**Technical Novelty And Significance:** 2
**Empirical Novelty And Significance:** 3
**Recommendation:** 5

**Clarity, Quality, Novelty And Reproducibility:**

As I noted above, the method is presented in a way that’s perhaps more formal and verbose than is actually necessary, especially since the diagonal assumption this paper makes dramatically simplifies the algorithm being proposed. Regardless, the paper is still fairly clear and understandable. I’m not aware of other previous works studying this topic, so in that sense the paper is original, though I don’t find the key findings to be all that surprising.

A Reproducibility Statement is included, and reproducing code is provided.


**Strength And Weaknesses:**

Strengths:
- S1. This paper’s topic is certainly interesting. Given only a few examples to learn from, the risk of overfitting to non-robust features is understandable. It’s well-known that few-shot accuracy on meta-test sets can vary wildly depending on which images/classes happen to be sampled, which is why the FSL community usually reports results averaged over thousands of meta-test sets. I’m not aware of any prior works that have specifically studied this problem.
- S2. Experiments do show a significant drop-off in performance with the worst case datasets produced by the proposed NMMD attack. This drop-off is reflected across several models and datasets, so this appears to be a general phenomenon rather than a quirk of a particular architecture or dataset. This is perhaps unsurprising, but Table 1 does clearly quantify how bad this drop can be.
- S3. The writing is fairly understandable. That said, the draft could use another round of proofreading, as there are a number of grammatical and idiomatic errors. See Miscellaneous below for a non-exhaustive list.

Weaknesses:
- W1. Terminology: I’m not confident that the framing of the methodology is quite right. I can see the connections and how the authors may have been inspired, but there are some subtle differences here that require care, as using terminology from other common problems can lead people to make assumptions about the setting and the proposed solution.
  - Adversarial attacks: Adversarial attacks typically refer to a scenario where an adversary trying to sabotage the model with inputs with imperceptible changes. There is no adversary here, nor are the changes “imperceptible,” as there are no modifications to individual images here. Rather, we’re specifically selecting the worst images for training.
  - Distribution shift: While the distribution here technically is indeed different and therefore a “shift,” it’s due to a specific sampling strategy, not because of factors that people usually refer to as distribution shifts (e.g. domain changes, temporal evolution).
  - Robustness: I would also take caution when talking about neural networks having poor “robustness”, which often refers to individual samples during inference, rather than training datasets.
  - Few-shot: The experimental evaluation is more akin to standard supervised training and evaluation in the low data regime, as opposed to the meta-dataset set-up common in FSL: FSL typically has both far fewer examples per class (e.g. 5, as opposed to 500 in this paper), and evaluation of transfer from base classes to novel classes (train and eval here are done on the same classes). These differences do not necessarily invalidate the results, but it may be a bit confusing to people expecting more typical FSL benchmarks like MiniImageNet/TieredImageNet when calling the method few-shot. NMMD seems like it should apply to such setting, so the authors may consider more standard FSL benchmark as well.

- W2. Methodology: If my understanding is correct, the method is just identifying the examples in each class that result in the largest magnitude gradients when the model is randomly initialized. This doesn’t strike me as necessarily resulting in the worst case few-shot sets. In particular, tossing the off-diagonal terms means inter-sample terms are discarded. This seems incongruous with the authors’ claim in the Introduction that lack of robustness can occur due to spurious correlations between samples. Furthermore, when I look at the resulting few-shot sets in Appendix G, these don’t strike me as being adversarially challenging few-shot sets, nor are spurious correlations visible. While it was the original goal, I don’t know if “worst” case is the right term here.

- W3. Notation: While I appreciate the desire to be precise, I do find some of mathematical notation somewhat unnecessarily verbose. For example, Section 4 uses several paragraphs largely to say that we assume there’s a model that generalizes from the support set to the query set (an underlying assumption in ML), that the support and query sets have the same label distribution (a standard property of classification benchmarks), and that we’re trying to find data samples that result in the highest error (already stated in the Introduction and Related Work).

- W4. Practicality: While the results are somewhat interesting, what’s the significance? When collecting a dataset, it’s highly unlikely that one would be unlucky enough to assemble such a worst case dataset, and one can mitigate the effects by collecting more data. As an adversarial attack perpetuated by a malicious actor, NMMD is not practical either, because the adversary would have to replace the entire dataset (in contrast to, say, data poisoning attacks), and the choice of including the hardest image samples would likely be highly noticeable to the model owner.


Questions:

- Q1. Given that popular datasets are known to have errors or ambiguity [1,2], I wonder if the worst case results in support sets with mislabeled support sets [3]? How do we ensure that worst case is still representative or useful?
- Q2. What were the pre-trained models pre-trained on? ImageNet? If so, there is a lot of overlap with the CIFAR-10 classes, so this isn’t truly a few-shot setting. These models would similarly be much less significantly impacted by a handful of training samples, which would explain the much smaller drop in performance from NMMD.
- Q3. How was the test accuracy evaluated? Was it the accuracy at the end of training, or did you use early stopping? Or was the best test accuracy during training reported? Asking because the train accuracies show strong evidence of overfitting (mostly 99, 100%) and 200 epochs as reported in Table 4 is a lot. When the test accuracy is reported may have a strong impact on the results.
- Q4. In few-shot learning evaluations, we often randomly sample not only the K shots per class, but also the N-way classification task. Have you considered how specific combinations of classes may result in extra poor performance?
- Q5. Have you considered looking at the reverse scenario? Which samples give the best performance? How are they different from the worst case samples? Can the best case samples be used a compact, more portable version of the dataset?


[1] Yun et al. Re-labeling ImageNet: from Single to Multi-Labels, from Global to Localized Labels. CVPR 2021.

[2] Northcutt et al. Pervasive Label Errors in Test Sets Destabilize Machine Learning Benchmarks. NeurIPS D&B 2021.

[3] Liang et al. Few-shot Learning with Noisy Labels. CVPR 2022.

Miscellaneous:
- “few-shot” vs “few shot” as a hyphenated adjective
- P_few isn’t really used and therefor doesn’t seem like a necessary notation to introduce.
- Section 4: “Assumption. 4.1”, “Assumption. 4.2” <= extra period
- Section 5: “Here, We first proves” <= “Here, we first prove”
- Section 5: “Last” <= “Lastly”
- Definition 5.1: notation collision. y is already used as the labels in Sec. 3 + 4.
- Theorem 5.1: “occurred” <= “incurred”?
- Equation 4: Should 1/m be 1/k and x_i_m be just x_m?
- Section 6: “a feed-forward neural networks” <= “a feed-forward neural network”. Also, all the models listed are feed-forward neural networks. The description of 1) relative to the other models needs to be clearer.
- Section 6: The distinction between models like ResNet and DenseNet with “pretrained models” ViT and EfficientNet is confusing. All of the listed models can be pre-trained.
- Section 6: notation collision for m, which is used as the number of trials and as an index in Eq 4.
- Section 7.1: Missing space after “3)”
- Table 3: “Average-case” is confusing. I initially thought this column was the average across the previous columns. Better call it “Random” to better match the text.
- Reproducibility Statement: What’s 6? “A” is “Appendix A”?


**Summary Of The Paper:**

This paper seeks to investigate the worst-case performance of neural network models in the low-data regime with respect to data sample index choice. Studying this problem is motivated by wanting to investigate neural network’s tendency to rely on spurious correlations for classification. The authors propose a method called NMMD attack that specifically selects the dataset subset that results in the worst performance. Experiments on several popular vision datasets (MNIST, CIFAR, ImageNet) and a variety of neural network architectures show that model performance does indeed suffer from picking the worst-case subset of the dataset.

**Summary Of The Review:**

Studying the worst case performance of models as a function of which subset of the data is selected is an interesting problem, and not one I’m aware of much prior work on. W2 and W4 are my main concerns. With the approximations that the authors made in order to achieve tractibiliy, the authors aren’t truly finding the “worst” case datasets, as is claimed. Even if they are though, it’s not clear to me what the practical use of the proposed method is: as an adversarial attack, it would fail to be particularly sneaky, and as a warning to ML practitioners, it would seem that the likelihood of picking such a bad dataset subset are relatively unlikely. If the goal is just to show that neural network performance drops if we try pick the training set giving the worst-case accuracy, then the findings are perhaps somewhat unsurprising.

---

> ### Author Response · Authors · 2022-11-08
> **We thank the reviewer for giving us an opportunity to explain the goal before concluding.**
>
> Our goal has two folds:
>
> 1. We aim to build a worst-case benchmark. Our results show worst-case evaluation is necessary to avoid "false high performance" evaluation under limited data.
>
> 2. We aim to find an easy-to-implement but effective method that can find a universal and challenging set to "attack" diverse models, which enables these sets to be a benchmark to evaluate worst-case performance given a new model. By combining the highest performance and worst-case performance together, we can better evaluate the learning ability of AI models.
>
> [5] Practicality concerns:
>
> We provide comprehensive benchmarks to evaluate the learning abilities of AI models.
>
> Learning over limited data is a hot research topic to evaluate the ability of AI models. We can see rapidly increasing benchmarks with limited data, like SuperGLUE. However, such benchmarks usually face the problem of "false high performance". With the increasing model capacity, more and more models claim that they are strong models with better test results under limited data settings. Does it mean that such models learn key features to handle all cases?
>
> Our work provides "challenging sets". The performance on these sets provides a new perspective to see the learning ability of an AI model.

---

> ### Author Response · Authors · 2022-11-08
> **We thank the reviewer for the constructive comments and suggestions.**
>
> [1] grammatical, notation, and idiomatic errors.
>
> We thank the reviewer for pointing out these presentation errors. We will carefully proofread the paper.
>
> [2] Terminology
>
> We thank the reviewer for pointing out these unclear points. We will add more comparisons between our work and related concepts. In addition, we will update the unclear points for better clarification.
>
> For adversarial attack, we will avoid using "attack" to describe our method. For distribution shift, we will use the more precious phrase "sampling shifts" to describe the change of the training set caused by the sampling policy. For robustness, we will use large empirical risk variance to replace "poor" robustness. For few-shot: we will use learning with limited data as a replacement.
>
> [3] identify the examples when the model is randomly initialized.
>
> We use approximation methods to estimate the worst-case sets, which is necessary since fine-tuned models as a backbone have the risk of data leak. If we use the fine-tuned model to select data, it is inevitable that the searched set overfits fine-tuned models and fine-tuned data. In this way, the searched set will lose its "attack" ability to other models.
>
> [4.1] Discarding the off-diagonal term
>
> Spurious features are statistical features that the model tends to overfit on when it comes in great abundance. Intuitively, the group of samples with larger gradient norms is learned less well and contains fewer features the model overfits on. Therefore, choosing a few-shot set according to gradient norm ranking has the potential to find subsets without spurious correlations.
>
> [4.2] The necessity of worst-case simplification.
>
> The main reason is the intractable search space. Our method is a simplified solution to estimate the worst-case performance, including approximation tricks like tossing the off-diagonal terms and assuming the stability of the NTK matrix. We agree that there is still exploration space to design more advanced solutions, though experiments show that the simplified solution works.
>
> [4.3] nor are spurious correlations visible on the searched set.
>
> Spurious correlation is hard to detect. We take a background example to see less spurious correlation in the search set. Figure 4 shows that the selected subset has more figures with white backgrounds, reducing the possibility that the model cheats with spurious background features.
>
> [5] Error examples
>
> First, we manually check the quality of NMMD-set sampled from CIFAR10 in Section 7. The visualization analysis is given in Appendix G.  We do not observe the overestimation of mislabeled data. All examples have the right labels. We also sampled 100 images from ImageNet. The same findings are observed.
>
> In addition to human check, similarity analysis also can provide a feasible solution to detect mislabeled data. If an example has an extremely high diversity score than other examples, it is likely to be a mislabelled example.
>
> [6] What were the pre-trained models pre-trained on?
>
> Since many previous studies have proven that pre-trained models are more effective and robust few-shot learners benefit from the large corpus, we also conduct these experiments.  Despite being pre-trained on ImageNet, these pre-trained models still have serious performance drop problems, showing the effectiveness of the proposed solution.
>
> [7] Test accuracy evaluated?
>
> The results are all the best test accuracy during training.
>
> [8] Have you considered looking at the reverse scenario?
>
> We agree that it is a good question to explore, though our paper focuses on a different topic. We will consider this question carefully.

---

> > ### Comment · Reviewer_di2U · 2022-11-08
> > **Response**
> >
> > Thank you for the response. I've read the other reviews and the authors responses, and I've been following the discussion in response to Reviewer Hiwt's review. While the initial review's score was a little harsher than I would give, there are many similarities in content with mine, and I largely agree with the sentiment. Overall, I largely concur with Reviewer Hiwt. While there is some interesting content here, the paper would strongly benefit from some re-framing and improved experiments, particularly the random baseline (more details below). The changes required may be too large to reasonable expect within the rebuttal period or for a camera-ready, so I keep my score. I'm hoping the authors take the feedback to improve the paper.
> >
> > ## Worst case
> > [4.2] I understood the intractability of finding the actual absolute worst case set from my initial reading of the paper. My concern is how good of an approximation the proposed method is, and how far from the actual worst case set is. In particular, I find response [7] a little concerning. Given that ImageNet does have wrong/ambiguous labels, I would expect that the method *should* recover such examples as part of the worst case set.
> >
> > In a similar vein, I also agree with the importance of the baseline that Reviewer Hiwt is asking for: minimum performance over $m$ sets drawn uniformly at random. The authors are right that they did in part perform this experiment, but I agree with Reviewer Hiwt that it isn't reported in the most meaningful way, and I disagree with the authors' assertion that variance gives a sense of minimum. While the mean and variance are useful to report, to me part of the importance of this baseline is to get a sense for how good the proposed method was at finding the worst case set compared to many random sets, especially in light of the approximations the authors made. As such, this random sets experiment is far more informative if the authors run many more trials and report the minimum; yes, it's intractable to compute every possible set, but the $m=3$ or $5$ quoted in the paper is too few.
> >
> > ## Goals
> > I thank the authors for their explanation, but the motivation is still a little unclear here. For goal #2, who would be performing this attack, and how? Why do existing benchmarks show "false high performance"? Can you point to an example of a benchmark with false high performance? Even for datasets with limited data, the dataset itself is fixed, so false high performance or worst case performance, there's not much that can be done besides expanding the dataset, which should at least partially alleviate any misfortune of picking a "worst case" set.
> >
> > Perhaps it's interesting to view this work from the angle of the importance of data quality? E.g. given a budget of 50 images, what the best and worst dataset you could sample? How are such datasets different?
> >
> > On spurious correlations: There's something somewhat backwards here about the evaluation. The paper is making the argument that relying on spurious correlations is bad, but the results clearly show that removing these spurious correlations results in worse performance. I can see this making sense: the test set probably contains the same spurious correlations. I think the authors can better prove their point though if they could identify a test set where relying on spurious correlations fails. Then, the "worst case" set would actually have some practical value in that it would generalize better to such a test set.
> >
> > ## Other
> > - Response [3] and [6] seem incongruous. If we're worried about data leakage from fine-tuning [3], then pre-trained models [6] will definitely also have this issue.

---

> > > ### Author Response · Authors · 2022-11-10
> > > **Random baseline and wrong labels**
> > >
> > > We thank the reviewer's suggestions and will carefully revise our paper following all reviewer's suggestions.
> > >
> > >
> > > [1] My concern is how good of an approximation the proposed method is, and how far from the actual worst case set is.
> > >
> > > We compare the empirical bound of random sampling and our method. We uniformly sample 50 sets from CIFAR10 and report the lowest performance in the following table.  By comparing this table with Table 1, we can see that the lowest performance is still much higher than NMMD-attack results.
> > >
> > > |          |  cifar10 |
> > > |----------| ---------|
> > > | ffn      |  47.22   |
> > > | vgg      |  64.72   |
> > > | resnet   |  60.69   |
> > > | resnext  |  59.58   |
> > > | densenet |  69.13   |
> > >
> > > Due to time limitations, we only sample 50 sets and will report more results in our paper.
> > >
> > >
> > > [2] In particular, I find response [7] a little concerning. Given that ImageNet does have wrong/ambiguous labels, I would expect that the method should recover such examples as part of the worst case set.
> > >
> > > Our optimization target (eq. 1) is based on high-quality dataset assumptions. The target is indeed to find the worst set.
> > >
> > > We also consider the wrong labels in our solution. We actually optimize Eq.4 (maximum intra-group similarity and minimum inter-group similarity). The maximum intra-group similarity can avoid the effects of wrong examples with random features.
> > >
> > > It is a good point and we will add more discussion about wrong labels to our paper.

---

> > > > ### Author Response · Authors · 2022-11-10
> > > > **Concerns about the goal**
> > > >
> > > > [3]  The motivation is still a little unclear here. For goal #2, who would be performing this attack, and how?
> > > >
> > > >
> > > > For model designers, this attack can be performed to evaluate new models. Worst-case optimization is a hot research direction.  Researchers can use it to evaluate the robustness of new models. They can directly evaluate models based on the challenging set.  For a new task, the researcher also can run our code to find the "attack" set and then evaluate new models.
> > > >
> > > > For researchers focusing on distributionally robust optimization, our attack provides a practical solution to evaluate whether these models can achieve high generalization ability to group shifts.
> > > >
> > > > We have compared the difference between the set with the worst performance and a random set.  The interesting point is that such differences are hard to identify by humans, as shown in Appendix. It is almost impossible for humans to build a high-quality dataset without spurious correlations. The dataset constructor can also use our method to find a high-quality set.
> > > >
> > > >
> > > > [4] Why do existing benchmarks show "false high performance"?
> > > >
> > > > To be precise, the main problem is that when testing the transfer ability of models on few-shot samples, most papers are based on random sampling.  We argue that this kind of comparison based on “chosen” samples is not fair enough as “cherry-picked” few-shot samples would result in “false high performance”.  Since there is a large gap between the random set and the performance on the set searched by our method, we question whether previous methods can be “false high performance".
> > > >
> > > > Current few-shot evaluations on large pre-trained models [1,2,3,4,5,6] often use the setting where the authors compare their results on the same large original dataset, e.g. GLUE, ImageNet, ILSVRC-2012, yet they use random sampled or hand-picked few-shot samples. This setting could be much improved by our proposed worst-case few-shot benchmark.
> > > >
> > > > Apart from work that fails to use standard benchmarks, there are papers that discuss few-shot standard benchmarks [7][8] for CV tasks, which take into consideration both cross-class generalization and intra-class differences (a general benchmark involving both meta-learning & transfer learning setting for few-shot evaluation). However, these benchmarks also involve random sampling for choosing samples within a class.
> > > >
> > > >
> > > >
> > > > Setting 1: “random collects K examples from each label for training”
> > > > NLP domain:
> > > > 1. Logan IV, Robert L., et al. "Cutting down on prompts and parameters: Simple few-shot learning with language models." arXiv preprint arXiv:2106.13353 (2021).
> > > > 2. Gao, Tianyu, Adam Fisch, and Danqi Chen. "Making pre-trained language models better few-shot learners." arXiv preprint arXiv:2012.15723 (2020).
> > > > 3. Sun, Tianxiang, et al. "Black-box tuning for language-model-as-a-service." arXiv preprint arXiv:2201.03514 (2022).
> > > > 4. Brown, Tom, et al. "Language models are few-shot learners." Advances in neural information processing systems 33 (2020): 1877-1901
> > > >
> > > > CV domain:
> > > > 1. Kolesnikov, Alexander, et al. "Big transfer (bit): General visual representation learning." European conference on computer vision. Springer, Cham, 2020.
> > > > 2. Alayrac, Jean-Baptiste, et al. "Flamingo: a visual language model for few-shot learning." arXiv preprint arXiv:2204.14198 (2022).
> > > >
> > > > Setting 2: split the dataset into train class, validation class, and test class.
> > > >
> > > > 1. Chen, Wei-Yu, et al. "A closer look at few-shot classification." arXiv preprint arXiv:1904.04232 (2019).
> > > > 2.  Dumoulin, Vincent, et al. "A unified few-shot classification benchmark to compare transfer and meta-learning approaches." Thirty-fifth Conference on Neural Information Processing Systems Datasets and Benchmarks Track (Round 1). 2021.
> > > >
> > > > [5] Perhaps it's interesting to view this work from the angle of the importance of data quality.
> > > >
> > > > It is indeed interesting to see what's the difference between the set with the worst performance and the set with the best performance.
> > > >
> > > > We did not exactly conduct these experiments. But we have compared the difference between the set with the worst performance and a random set.  The interesting point is that such differences are hard to identify by humans, as shown in Appendix.
> > > >
> > > >
> > > > [6] If we're worried about data leakage from fine-tuning [3], then pre-trained models [6] will definitely also have this issue.
> > > >
> > > > There are two different kinds of data leakage.
> > > >
> > > > We worried about data leakage when generating the "attack" set based on a fine-tuned model.  It will overfit to fine-tuned data and fine-tuned model. The generated set will lose its "attack" ability for diverse models.
> > > >
> > > > We do not adopt pre-trained models to generate the "attack" set. If there is overlapped data between ImageNet and CIFRA10, it will contribute to better average performance, but not ensure a smaller gap between average performance and the worst performance.

---

> > > > > ### Comment · Reviewer_di2U · 2022-11-17
> > > > > **Response**
> > > > >
> > > > > ## Minimum of random draws
> > > > > I thank the authors for being responsive, and for running the additional experiments. Yes, I do believe that reporting such a experiment showing the minimum is much more compelling baseline than what's currently reported in the paper. It does indeed appear that the proposed method results in a lower accuracy than any of the randomly drawn ones, but it also provides further evidence that the chances of being unlucky enough to draw such a bad dataset in practice seem slim (W4), which again leads me to question what the practical value is.
> > > > >
> > > > > ## Motivation
> > > > > I'm still not fully convinced of the motivation. In practice, when would we encounter such a worst case set? As I pointed out above, the chances of naturally encountering such a dataset is vanishingly small, so it's not clear to me why we should be optimizing for it.
> > > > >
> > > > > There's additionally some conflation in this argument. I don't see why this method necessarily produces a "high-quality set". Nothing about the objective ensures the set is "high-quality", only that it's a set that makes the model perform poorly. There are many reasons why this may be the worst case set, not just the removal of spurious correlations. It could in fact be picking poor examples. By seeking to avoid any spurious correlation, this method may in fact be inducing a domain shift relative to the test set, when ultimately we actually want a model that does the best on the test distribution.
> > > > >
> > > > > ## "False high performance"
> > > > > I disagree with the assertion that most papers are "cherry picking" few-shot samples during their evaluations, leading to "false high performance". I find several problems with this argument:
> > > > > 1. There is indeed high variance in accuracy based on the sampled meta-test task in FSL papers (though a large part of that variance is in part due to the sampling of test classes, which does not play a role in this paper's experimental setting since the train and test classes are the same), but these works often mitigate this variance by evaluating on *many* such meta-test tasks, sometimes up to 10K in certain FSL papers, at which point the random luck factor is mostly smoothed out.
> > > > > 2. Unless the authors have reproduced any of the results in these past papers and can show that the reported results that are statistically significantly higher than the mean of many random draws, then it's somewhat irresponsible to accuse these works of cherry picking. Please provide concrete empirical examples if this is going to be a motivating claim.
> > > > > 3. Because this paper is lacking any experiments showing how much better than average a "best-case" dataset is, it's additionally unclear if "false high performance" is really that much higher than expectation.
> > > > >
> > > > > ## Miscellaneous
> > > > > Not factoring this into my decision or score, as the paper is quite recent and I didn't bring it up in my initial review, but the authors may find it interesting to discuss the work "Beyond neural scaling laws: beating power law scaling via data pruning" (https://arxiv.org/abs/2206.14486).

---

### Official Review · Reviewer_X5Fg · 2022-10-25

**Confidence:** 4
**Correctness:** 3
**Technical Novelty And Significance:** 3
**Empirical Novelty And Significance:** 3
**Recommendation:** 5

**Clarity, Quality, Novelty And Reproducibility:**

The paper is clearly written. The theoretical statements have some issues, which I have pointed out above. However, the experiments are novel and clearly show the fragility of models in few-shot settings.

**Strength And Weaknesses:**

Strengths:

The paper conducts multiple experiments to show that for a randomly initialized or a  pretrained model we can form a small subset of examples with the largest gradients at initialization (not necessarily of the same model), on which the model's performance drops drastically, compared to a random subset of examples. This clearly shows the fragility of different models in the few-shot setting. Furthermore, with a small ablation study, the authors show that the "adversarial" subset discovered is as diverse as a random subset of examples, but doesn't contain much spurious correlation that the model can latch on. This further shows the fragility of the real-world models to latch on to spurious correlations to perform well in few-shot settings.


I have the following questions/concerns:

a) In eq. (2), the difference between $\epsilon_Q( f_{I_k} ) $ and $\epsilon_Q (f)$ is given by the MMD( P_{I_k}, P ). However, in theorem 5.2, we approximate $ MMD_f ( P, P_{I_k} ) $ with the NTK w.r.t. the function $f$. How do the authors relate the difference between $\epsilon_Q( f_{I_k} ) $ and $\epsilon_Q (f)$ with NTK w.r.t. $f$?

b) Does the NTK assumption (the kernel doesn't change much during training) hold true for the models in the few-shot training in Table 1?

c) What is the empirical gap observed in eq. (3) using the NTK of the models in Table 1? Does the NTK estimate give a non-trivial empirical bound for $\epsilon_Q( f_{I_k} ) - \epsilon_Q(f)$ in equation (3)?

d) How do VGG-attack and ResNeXt-attack perform in table 2? Will we observe a similar drop in performance?

Furthermore, how do subsets using gradients of ViT-B/16  and EfficientNetV2 perform in Table 2? Should one expect the drop in performance to be higher than the ones with the FFN attack?

e) If the NTK assumption holds true, then we can also use the NTK of a few-shot finetuned model to gather the "adversarial" subset.

For the models used in Table 1, how do the models perform against the NMMD attack from their finetuned versions? That is if we finetune a model on a small random subset and use the model to create the adversarial subset, will we observe a similar drop in performance?  What about an NMMD attack with respect to the gradients of a model that has been finetuned on the entire dataset?



**Summary Of The Paper:**

The paper aims to evaluate the few-shot capabilities of a neural network on the worst-case subset of a dataset. The main motivation is to showcase the fragile ability of neural networks to memorize spurious statistical cues in the dataset, leading to poor generalization. To find the worst-case subset, the authors propose an algorithm that picks the subset with the highest gradient norms at initialization. With extensive experiments, they showcase the failure of well-known models in fewshot setting on datasets like CIFAR-10, CIFAR-100, and Imagenet-1k.

**Summary Of The Review:**

Overall, my scores are borderline. The paper proposes a novel attack to understand the relation between few-shot evaluation and spurious correlations between training and test sets. However, there are issues with the theoretical motivation underlying the proposed algorithm. Hence, I would like the authors to clarify the questions raised above.

---

> ### Author Response · Authors · 2022-11-10
> **We thank the reviewer for the constructive comments. The followings are our point-to-point responses.**
>
>
>
> [1]  In eq. (2), the difference between ϵQ(fIk) and ϵQ(f) is given by the MMD( P_{I_k}, P ). However, in theorem 5.2, we approximate MMDf(P,PIk) with the NTK w.r.t. the function f. How do the authors relate the difference between ϵQ(fIk) and ϵQ(f) with NTK w.r.t. f?
>
> There may be some misunderstandings. We don't approximate the ϵQ(fIk) and ϵQ(f) terms, as they are directly accessible as the test loss of the original model. We only approximate the MMD( P_{I_k}, P ) term during computation, as directly calculating MMD would be intractable.
>
> [2] Does the NTK assumption (the kernel doesn't change much during training) hold true for the models in the few-shot training in Table 1?
>
> Although the NTK kernel assumption has been theoretically proven in paper [1], we still admit that there is a gap between empirical value and theoretical value.
>
> The reason for choosing randomly-initialized models, rather than fine-tuned models, is to avoid the data leakage problem in fine-tune version. If we use fine-tuned models to select data, it is inevitable that the searched set overfits fine-tuned models and fine-tuned data. In this way, the searched set will lose its "attack" ability to other models.
>
>
> 1. Sanjeev Arora, Simon S Du, Wei Hu, Zhiyuan Li, Russ R Salakhutdinov, and Ruosong Wang. On
> exact computation with an infinitely wide neural net. Proc. of NeurIPS, 2019
>
> [3]  What is the empirical gap observed in eq. (3) using the NTK of the models in Table 1? Does the NTK estimate give a non-trivial empirical bound for ϵQ(fIk)−ϵQ(f) in equation (3)?
>
> First as stated in reply to question 1, we didn't approximate the ϵQ(fIk)−ϵQ(f). ϵQ(fIk) is the test loss of the model trained on the searched subset. ϵQ(f) is the test loss of the model trained on the entire training set.
>
> As for the empirical gap in eq.(3), getting the empirical gap would be quite intractable, as the constant \epsilon_H is related to the complexity of hypothesis space, and is intractable to estimate in deep learning. Since it is intractable, previous related studies [1][2][3]  do not calculate the empirical gap either.
>
> Since it is hard to estimate, to compare the empirical bound of random approximation and our methods, we randomly sample 50 sub-sets from CIFAR10 and report their lowest performance in the following table.  As we can see, the lowest performance is much higher than NMMD-attack results.
>
> |          |  cifar10 |
> |----------| ---------|
> | ffn      |  47.22   |
> | vgg      |  64.72   |
> | resnet   |  60.69   |
> | resnext  |  59.58   |
> | densenet |  69.13   |
>
>
> 1. Peng, Xingchao, et al. "Moment matching for multi-source domain adaptation." Proceedings of the IEEE/CVF international conference on computer vision. 2019. (cited by 743)
> 2.  Ben-David, Shai, et al. "A theory of learning from different domains." Machine learning 79.1 (2010): 151-175. (cited by 2553)
> 3. Acuna, David, et al. "f-domain adversarial learning: Theory and algorithms." International Conference on Machine Learning. PMLR, 2021. (cited by 22)
>
> [4] How do VGG-attack and ResNeXt-attack perform in table 2? Will we observe a similar drop in performance?
>
> The results are shown in the following table. We observe a similar drop.  We will add these results to our paper.
>
> |                | cifar10(500) ResNeXt | cifar10(50) ResNeXt | cifar10(500) vgg | cifar10(50) vgg |
> |----------------|----------------------|---------------------|------------------|-----------------|
> | ViT-B/16       | 96.44+0.11           | 70.50+3.06          | 97.12+0.10       | 87.28+1.11      |
> | EffientNetV2-S | 89.67+0.31           | 47.23+3.53          | 91.62+0.25       | 68.55+2.64      |
>
> [5] Furthermore, how do subsets using gradients of ViT-B/16 and EfficientNetV2 perform in Table 2?
>
> As described in response #4, we find that models with worse performance as the backbone can generate more "challenging" sets.
>
> Furthermore, these models are pre-trained on ImageNet. As suggested by Reviewer #2, there are overlapped images between ImageNet set and CIFAR10 set. If we choose these pre-trained models to select "attack" set on CIFAR10, it will overfit the overlapped set.
>
>
> [6] What about an NMMD attack with respect to the gradients of a model that has been fine-tuned on the entire dataset?
>
> The following table shows our results by using fully fine-tuned data on the entire dataset. The "attack" performance is worse than the model based on randomly-initialized models in Table 1.
>
> |                 |  cifar10          |
> | ------------- | ----------------  |
> | ffn            |  46.62+0.54   |
> | vgg          |  58.91+1.04   |
> | resnet      |  58.30+0.85   |
> | resnext    |  59.42+1.22   |
>
> Using fine-tuned models has the risk of data leakage. Please see response #2.

---

### Decision · Program_Chairs · 2023-01-20

**Decision:**

Reject

**Justification For Why Not Higher Score:**

All reviewers found that this paper did not meet the bar for ICLR. Two revewers engaged in extensive discussion, raising issues with the framing of the paper, the thoroughness of the experiments, and the utility of the method. The authors rebuttal to these points was considered but in the end the AC agrees the paper needs more work. It could help to add more experiments on methods specifically designed for few-shot / limited data settings, and it also might help to demonstrate practical uses, such as using the discovered worst case sets to improve model robustness.

**Justification For Why Not Lower Score:**

N/A

**Metareview: Summary, Strengths And Weaknesses:**

Summary:
This paper studies the robustness of few-shot learners to adversarially selected training sets. The paper introduces an algorithm to approximate worst set selection by searching for the label-balacned subset of images with largest (approximate) MMD w.r.t. the full training set. This attack reduces the performance of a variety of few-shot learning methods.

Strengths:
* Interesting novel problem
* Good analysis of spurious features in selected set

Weaknesses:
* Practical utility is unclear
* The approximation to MMD makes strong assumptions and it's unclear if it is a good approximation
* The framing of the paper could use work; in particular, the community might not consider this to be few-shot (50-500 shots), and the paper mostly does not evaluate algorithms specifically designed to be few-shot learners (the proposal to reframe as "limited data" may help)
* Generally, the reviewers did not find the results to yield clear and significant insights